# Recollection in the human hippocampal-entorhinal cell circuitry

Bernhard P. Staresina[1,2], Thomas P. Reber [3,4], Johannes Niediek [3], Jan Boström[5], Christian E. Elger [3] &
Florian Mormann [3]

Imagine how flicking through your photo album and seeing a picture of a beach sunset brings back fond memories of a tasty cocktail you had that night. Computational models suggest that upon receiving a partial memory cue ('beach'), neurons in the hippocampus coordinate reinstatement of associated memories ('cocktail') in cortical target sites. Here, using human single neuron recordings, we show that hippocampal firing rates are elevated from ~ 500–1500 ms after cue onset during successful associative retrieval. Concurrently, the retrieved target object can be decoded from population spike patterns in adjacent entorhinal cortex (EC), with hippocampal firing preceding EC spikes and predicting the fidelity of EC object reinstatement. Prior to orchestrating reinstatement, a separate population of hippocampal neurons distinguishes different scene cues (buildings vs. landscapes). These results elucidate the hippocampal-entorhinal circuit dynamics for memory recall and reconcile disparate views on the role of the hippocampus in scene processing vs. associative memory.

[1] School of Psychology, University of Birmingham, Birmingham B15 2TT, UK. [2] Centre for Human Brain Health, University of Birmingham, Birmingham B15 2TT, UK. [3] Deptartment of Epileptology, University of Bonn Medical Centre, Bonn D- 53175, Germany. [4] Faculty of Psychology, Swiss Distance Learning University, Brig 3900, Switzerland. [5] Department of Neurosurgery, University of Bonn Medical Centre, Bonn D- 53175, Germany. Correspondence and requests for materials should be addressed to B.P.S. (email: b.staresina@bham.ac.uk)

Memory is central to adaptive behaviour across species[1]. Despite concerted efforts, however, it is still unclear how we are able to retrieve a rich memory trace of past experiences after receiving a simple reminder cue ('episodic memory'). Converging evidence points to a critical role of the medial temporal lobe (MTL), most notably the hippocampus, for intact episodic memory[2]. Based on the seminal discovery of hippocampal place cells coordinating spatial navigation in rodents[3,4], one prominent view of the human hippocampus emphasizes a dedicated role in spatial/scene perception and scene construction[5–7]. Classic computational models and human neuropsychological work, on the other hand, have highlighted mnemonic operations such as associative/relational memory formation and retrieval as well as pattern completion[8–11]. Pattern completion denotes a process through which hippocampal 'index cells'—upon receiving a partial memory cue—coordinate reinstatement of mnemonic representations in cortical target sites, particularly in entorhinal cortex (EC), the first cortical recipient of hippocampal output[12,13]. While constituting the backbone of episodic memory theory, evidence for a role of hippocampal neurons in coordinating cortical reinstatement during associative memory retrieval has been lacking. Some rodent studies employed simultaneous recordings from hippocampus and EC, but primarily focused on these regions' respective contributions to spatial navigation[14,15]. Owing to technical challenges pertaining to multi-site recordings, nonhuman primate electrophysiological recordings have focused either on the hippocampus or EC in isolation, without examining cross-regional dynamics[16–18]. Moreover, the question remains whether the human capacity of single-trial learning and the phenomenology of vivid recollection can easily be translated to experimental paradigms in these species. Conversely, human whole-brain neuroimaging only allows for indirect measures of neuronal activity, with spatiotemporal imprecision further impeding detection of fine-tuned hippocampal-cortical dynamics.

Here we capitalised on the rare opportunity to record action potentials from individual neurons in the human hippocampus and EC while participants ($n = 16$) performed alternating blocks of associative and nonassociative memory tasks (Fig. 1). In the nonassociative memory (NAM) task, participants saw scene images depicting buildings or landscapes during the study phase (encoding). During the test phase (retrieval), the same scene images were intermixed with novel scenes, and participants indicated whether or not they had seen the image during encoding, yielding two conditions of interest: HIT (correct identification of an old image) and Correct Rejection (CR; correct identification of a new image). In the associative memory (AM) task, participants again saw trial-unique scene images from the same building/landscape set during encoding. Superimposed on the given scene was a grey square including one of two object images. The choice of object images was customized for each experimental session based on a preceding screening session to identify response-eliciting stimuli (see Methods). At retrieval, participants saw the same scene images without the objects and indicated which of the two objects had been paired with the given scene, yielding the memory outcomes of correct Associative Memory (AM+) and incorrect as well as "don't know" responses (the latter two referred to as AM−).

We show that hippocampal firing rates increase shortly after cue presentation in both the nonassociative scene recognition task and the associative scene-object recall task. Importantly, firing rates are sustained from ~ 500 to 1500 ms for successful associative retrieval (AM+) only. At the same time, the successfully retrieved target object can be decoded from population spike patterns in adjacent entorhinal cortex (EC). Of note, individual hippocampal spikes precede EC spikes during AM+ and the rate of hippocampal firing predicts the strength of object reinstatement in EC. Critically, before orchestrating object reinstatement in EC, a separate population of hippocampal neurons distinguishes different scene cue types (buildings vs. landscapes) in both tasks (NAM and AM). These results reconcile competing models of hippocampal function (scene processing vs. associative memory) and elucidate the hippocampal-entorhinal circuit dynamics in service of human memory recall.

## Results

**Behavioural Results**. In the nonassociative memory (NAM) task, participants correctly recognized old images for 89% of trials (±2% SEM, HITs) and correctly identified new images for 90% of trials (±2% SEM, CRs). In the associative memory task, participants remembered the correct target object for an average of 79% of trials (±3% SEM, AM+), indicated the wrong target object for 20% (±3% SEM) and indicated they did not know the answer on the remaining 1% of trials (AM−). For both tasks, the HIT minus FA (NAM) and Correct minus Incorrect (AM) rate was significantly greater than zero (both $t_{(15)} > 11$, $P < .001$), confirming high retrieval accuracies in both tasks.

**Hippocampal engagement during memory retrieval**. Our first analysis focused on the hippocampus, where we recorded from 238 neurons in the anterior portion across the 16 participants (Fig. 2a; $M \pm SEM$ per participant: $14.9 \pm 2.6$ neurons). To determine whether and when hippocampal neurons might show preferential engagement for associative vs. nonassociative memory retrieval, we compared time-resolved spike rates for AM+ (associative) vs. HITs and CRs (nonassociative). As shown in Fig. 2b, hippocampal firing rates rapidly increased after ~200 ms post stimulus onset, with a peak at ~500 ms. Critically, while firing rates returned to baseline levels for NAM HITs and CRs thereafter, AM+ trials showed sustained levels of firing until ~1500 ms. A time-resolved one-way repeated-measures ANOVA with the factor condition (AM+, HITs, CRs) revealed a significant effect from 900 ms–1300 ms post cue onset ($P_{cluster} = .010$, corrected for multiple comparisons across time[19]), which subsidiary contrasts confirmed to reflect increased firing rates for AM+ compared to both HITs and CRs (both $t_{(237)} > = 2.85$, $P < = .005$), with no difference between the latter two ($t_{(237)} = 0.46$, $P = .637$) (Fig. 2b, inset). Within the same time window, we observed a relative increase in hippocampal firing rates for AM+ vs. AM−, unfolding at ~700 ms after cue onset ($P_{cluster} = .002$, Fig. 2c). These results pinpoint a hippocampal process specifically deployed for associative retrieval (above and beyond scene recognition) emerging at ~500 ms post cue onset. Figure 2d shows a single neuron example of raw spike trains for AM+ and NAM HIT trials and the corresponding time courses resulting from their convolution. Memory results for EC are shown in Supplementary Figure 1.

**Memory-driven object reinstatement in entorhinal cortex**. Models of pattern completion predict that the hippocampus coordinates reinstatement of memory content in cortical sites via EC[9,12,13]. We thus examined whether the identity of the successfully retrieved target object could be decoded based on population codes in EC, where we recorded from 211 neurons across our 16 participants ($M \pm SEM$ per participant: $13.2 \pm 1.6$ neurons). EC has previously been linked to object representations in rodents[20] and humans[21], particularly in its anterior/lateral portion from which we recorded here (Fig. 3a).

In a first step, we confirmed that object identity could be reliably decoded from EC population codes during learning. For each participant, we trained a linear discriminant analysis (LDA)

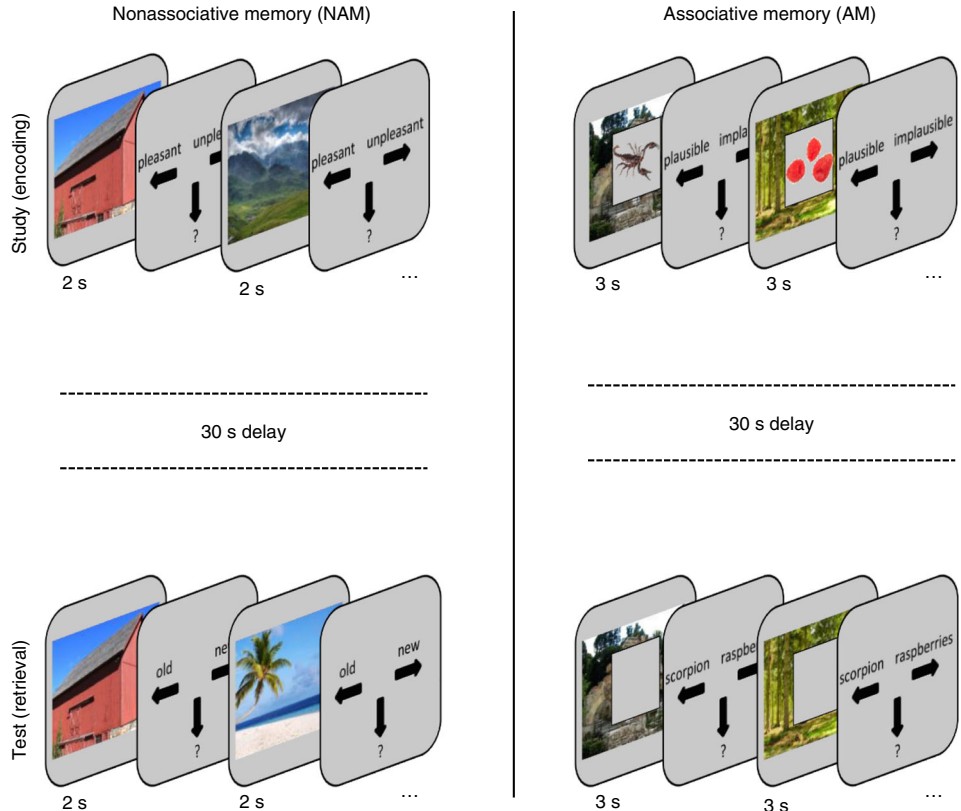

**Fig. 1** Experimental Paradigm. Participants performed alternating runs of nonassociative memory (NAM) and associative memory (AM) tasks. Each run consisted of a study (encoding), delay (30 sec) and test (retrieval) phase. Each NAM encoding phase included 20 scene image trials and each NAM retrieval phase included 40 scene image trials (50% old, 50% new). For AM runs, encoding and retrieval included 10 scene-object trials. We presented four NAM blocks and eight AM blocks (4x NAM-AM-AM), resulting in a balanced number of 80 old NAM trials, 80 new NAM trials and 80 AM trials during retrieval. AM tasks always included two objects, but exemplars varied across participants. Scene images obtained from https://commons. wikimedia.org under the CC-BY-SA-3.0 license

classifier to distinguish between the two objects. The feature vectors consisted of the activity of a given participant's EC neurons in response to each of the two objects, which was first averaged across the entire 3 s stimulus presentation and z-scored across trials. Decoding performance was derived via five-fold cross-validation and statistically evaluated against surrogate decoding performance, created by shuffling the training labels 100 times and averaging the resulting performance values to provide, for each participant, a single baseline value under the null hypothesis of label exchangeability[22]. Results showed decoding performance significantly exceeding chance by 12.94% (SEM = 3.58%) across participants ($t_{(15)} = 3.62$, $P = .003$ vs. surrogates). This indicates that EC population codes were indeed able to distinguish the two object stimuli in our paradigm. Note that no object decodability was observed in hippocampus in this paradigm ($t_{(15)} = 0.96$, $P = .350$ vs. surrogates; Supplementary Figure 2 and Discussion).

Next, we assessed whether and when encoding-related population codes might get reinstated during successful associative retrieval. To this end, we again trained the LDA classifier to discriminate between object 1 and object 2 based on all AM encoding trials, but did so in a time-resolved manner and then applied the training parameters to the AM+ retrieval data (where objects were retrieved but not shown), again in a time-resolved manner. Specifically, for both encoding and retrieval data, a sliding window of 500 ms was moved across the trial in 10 ms steps from −.5 to 3 s relative to stimulus onset. Within each

window, spike trains were averaged across time and z-scored across trials. The resulting time-by-time matrix thus indicates whether successful retrieval reinstates object encoding patterns and if so, which encoding time windows are most readily reinstated. As shown in Fig. 3b, results revealed evidence for reinstatement of the target object from ~600 to 1500 ms post stimulus onset during AM+ retrieval, where encoding patterns from ~1000 to 2000 ms were reinstated. Importantly, this reinstatement effect was not only significantly greater than chance performance (tested against surrogates as described above), but also significantly greater than AM− reinstatement (Fig. 3b), thus linking reinstatement to memory behaviour. Again, no such reinstatement effects were observed in hippocampus.

It deserves explicit mention that the reinstatement analysis described thus far performed decoding in each participant individually and then assessed the reliability of decodability across participants. However, we also sought to obtain converging evidence for target object reinstatement across all neurons (i.e. pooling across participants). To this end, the discriminability of object 1 vs. object 2 in a given neuron during AM+ retrieval was compared to the discriminability of object 1 vs. object 2 in the same neuron during all AM encoding trials. Discriminability was quantified as the t statistic resulting from comparison of all object 1 trials vs. all object 2 trials for a given neuron (again with the averaged 0–3 sec spike trains per trial as dependent variables). Results revealed a significant positive correlation for AM+ trials (Spearman $r = .19$, $P = .006$; Fig. 3c), indicating that across

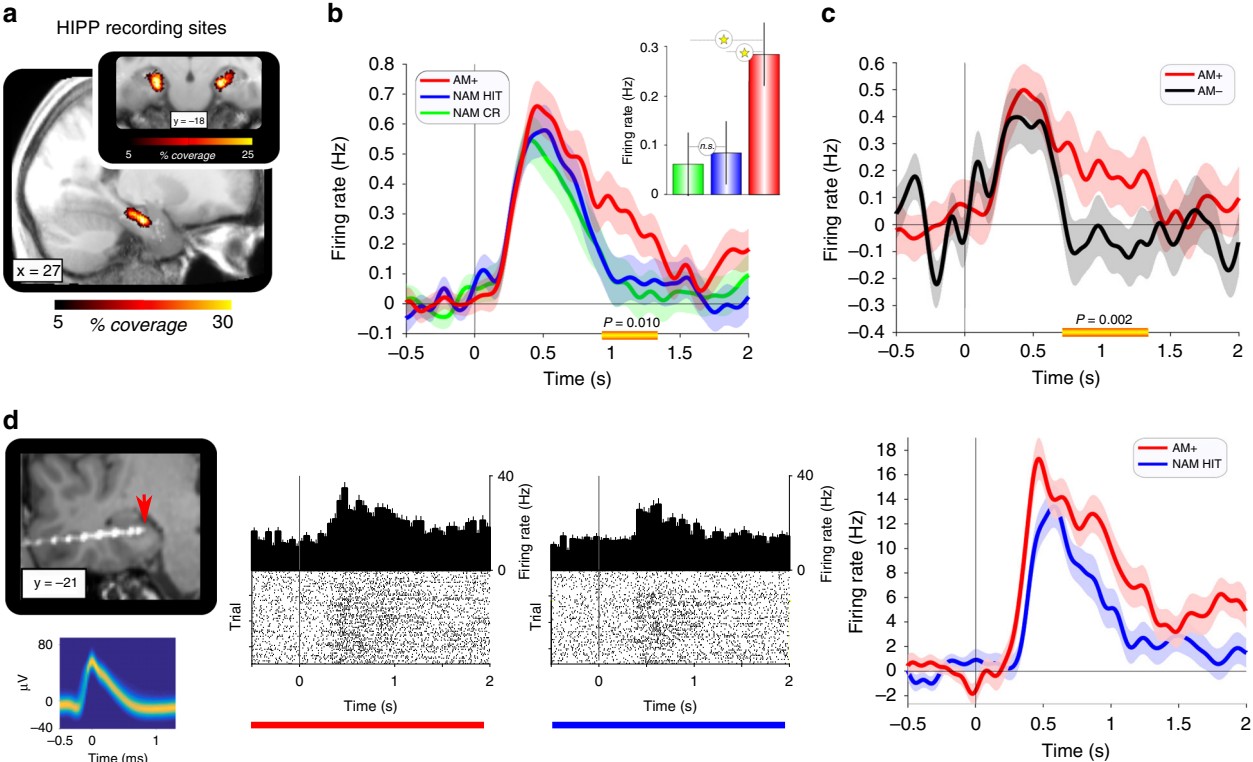

**Fig. 2** Hippocampal engagement during successful associative memory retrieval (AM+). **a** Hippocampal (HIPP) recording sites across participants projected onto the mean anatomical scan, normalized to MNI space. **b** Mean HIPP firing rates across neurons ($n = 238$, baseline-corrected) during successful associative retrieval (AM+), successful recognition of old images (NAM HIT) and successful identification of new images (NAM CR). Shaded areas show average standard error of all pairwise condition differences. Horizontal yellow line indicates a significant cluster of repeated-measures ANOVA Condition effects from 900–1300 ms post stimulus onset (corrected for multiple comparisons across the entire 0–2000 ms time window; Methods). The cluster P value is .010 and the maximum effect size ($F_{(2,474)}$) is 6.37 at 1000 ms. Inset shows results from pairwise follow-up contrasts, revealing a significant increase in firing rates for AM+ compared to both nonassociative conditions. **c** Mean HIPP firing rates across neurons ($n = 170$, after removing participants with less than 10 AM- trials) during successful vs. unsuccessful associative retrieval (AM+ vs. AM−). Shaded areas show standard error of the condition difference. Horizontal yellow line denotes a significant cluster of paired-samples $t$-test effects from 700–1300 ms post stimulus onset (corrected for multiple comparisons across the entire 0–2000 ms time window). The cluster P value is .001 and the maximum effect size ($t_{(169)}$) is 3.19 at 780 ms. **d** Single neuron example. left: participant's electrode placement, illustrated as a post-operative CT scan co-registered to the pre-operative MRI scan and normalised to MNI space. Arrow head indicates protruding microwires. Waveforms of action potentials are depicted as temperature-scaled density plots. middle: graphs show spike raster plots and peri-stimulus time histograms (PSTH, 50 ms bins) for AM+ (red) and NAM HITs (blue). right: corresponding time courses (mean ± SEM across trials) after convolving the spike trains with a Gaussian kernel (50 ms width) and subtracting the −.5 to 0 s baseline interval

neurons, a neuron that shows a preference for one object over the other at encoding tends to show the same preference during retrieval of the two objects. Note that no such correlation was seen for AM− trials (Spearman $r = .02$, $P = .787$; Supplementary Figure 3b), again linking reinstatement to memory behaviour. Figure 3d illustrates this target object reinstatement in a single example EC neuron.

**Relationship between hippocampal and EC retrieval effects.** So far, we have described that in a time window of ~500–1500 ms after cue onset, hippocampal firing rates increase during AM+ and, in parallel, the retrieved object identity could be decoded from EC population patterns. We next sought more direct evidence for a role of the hippocampus in coordinating EC reinstatement, as proposed by extant models of hippocampus-mediated pattern completion in cortical target sites[12,13]. First, we asked whether hippocampal firing rates and EC reinstatement might be directly related. To this end, we divided each participant's AM+ trials into high hippocampal firing trials and low

hippocampal firing trials based on a median split (after averaging across all hippocampal neurons in a given participant and across the 500–1500 ms time window). Next, we separated the trial-wise EC reinstatement values (% decoding accuracy, averaged across the 1–2 s encoding window and the .6–1.5 s retrieval window, Fig. 3b) based on trials with high vs. low hippocampal firing rates and compared the resulting values across participants via a paired-samples $t$-test. Indeed, results showed significantly greater EC reinstatement for AM+ trials with relatively high compared to relatively low hippocampal firing rates ($t_{(14)} = 3.27$, $P = .006$; Fig. 4a; one participant was excluded due to insufficient spiking across trials for a median split).

Second, if hippocampal firing orchestrates EC activity during AM+, there should be a temporal precedence of hippocampal relative to EC spikes. To assess such temporal precedence, we conducted a cross-correlogram (CCG) analysis, examining the precise timing of firing across hippocampus/EC neuron pairs. In particular, target spikes in individual EC neurons were temporally aligned with respect to seed spikes in individual hippocampal neurons. This was done for all possible same-hemisphere

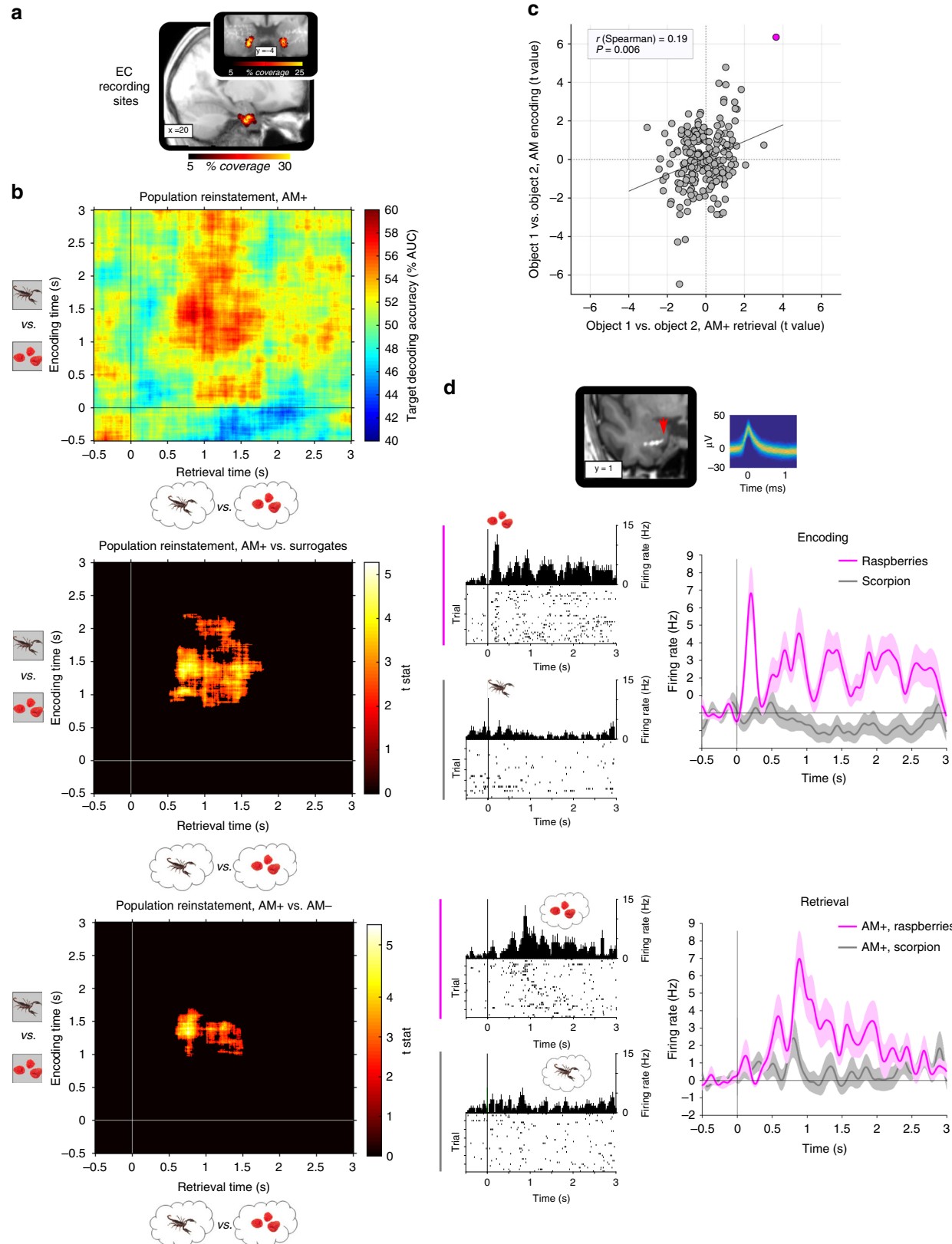

combinations of hippocampus/EC neurons in each participant, including spikes across the entire 0–3 s retrieval period. To control for trial-invariant firing properties across neurons, a 'shift-predictor' baseline CCG was derived by scrambling trial orders such that seed spikes came from trial n and target spikes from trial n ± 1. Relative increases in real- vs. shift-predictor CCG signify trial-specific coupling between two neurons, with potential asymmetries further informing about the underlying

**Fig. 3** Memory-driven target object reinstatement in entorhinal cortex (EC). **a** EC recording sites across participants projected onto the mean anatomical scan (MNI space). **b** Reinstatement across participants, encoding (rows) x retrieval (columns) time. top: Decoding accuracies (% area under the curve, AUC) for successful associative memory (AM+, maximum decoding accuracy of 59.41% averaged across participants). middle: AM+ vs. surrogates, revealing a significant cluster of paired-samples $t$-test effects from 600–1500 ms post stimulus onset during AM+ retrieval, reinstating ~1000–2000 ms encoding patterns (corrected for multiple comparisons across the from −.5 to 3 s encoding/retrieval time windows). Cluster $P$ value: .003. bottom: AM+ vs. AM−, revealing a significant cluster of paired-samples $t$-test effects from 600–1500 ms post stimulus onset, more strongly reinstating ~1000–1500 ms encoding patterns during AM+ (corrected for multiple comparisons across the from −.5 to 3 s encoding/retrieval time windows). Cluster $P$ value: .035. **c** Target object reinstatement across EC neurons. Scatter plot shows (**i**) discrimination of object 1 vs. object 2 during AM+ retrieval (x-axis, independent samples $t$-statistics based on the average firing rates across the entire 3 s retrieval period) against (**ii**) discrimination of object 1 vs. object 2 perception during all AM encoding trials (y-axis, independent samples $t$-statistics based on the average firing rates across the entire 3 s encoding period). Results show a significant positive correlation (Spearman $r = .19$, $P = .006$). **d** Target object reinstatement in a single EC neuron (magenta circle in the scatter plot). Top: participant's electrode placement (post-operative CT scan co-registered to the pre-operative MRI, MNI space). Arrow head indicates protruding microwires. Waveforms of action potentials are depicted as temperature-scaled density plot. middle/bottom: Left graphs show spike raster plots and peri-stimulus time histograms (PSTH; 50 ms bins), right graphs show the corresponding time courses (mean ± SEM across trials) after convolving the spike trains with a Gaussian kernel (50 ms width) and subtracting the from −.5 to 0 s baseline interval. This EC neuron shows selectivity for 'raspberries' vs. 'scorpion' during initial encoding and again during AM+ retrieval, with a delay in firing onset from encoding to retrieval

directionality while ruling out effects on two neurons caused by the onset of a stimulus or repetitive task events[23]. As shown in Fig. 4b, the resulting CCG revealed strong evidence for functional coupling between hippocampus and EC during AM+ retrieval. Critically, significant increases in EC firing were seen within the first 30 ms after hippocampal spikes ($P_{cluster} = .001$), with greater EC firing rates following than preceding hippocampal spikes ($t_{(1797)} = 3.25$, $P = .001$; ± 50 ms). No increase in EC firing with respect to hippocampal spikes was seen for AM− trials. In fact, there was a significant difference in EC firing following hippocampal spikes for AM+ compared to AM− trials ($P_{cluster} = .001$). Together, these findings reveal that firing in the human hippocampus precedes firing in EC target sites where mnemonic representations are reinstated during successful associative retrieval. While this set of results is consistent with a role of hippocampus coordinating pattern completion in EC, we note that evidence for causality would require an experimental intervention approach.

**Scene processing precedes associative retrieval in hippocampus.** Finally, what might be the functional significance of the early rise in hippocampal firing rates seen for both nonassociative and associative retrieval (starting ~200 ms post stimulus onset; Fig. 2b)? A substantial body of work points to a dedicated role of the hippocampus in spatial/scene processing[5–7]. We thus capitalised on the fact that we used two types of scene images (buildings and landscapes), testing whether these scene types could be decoded based on firing patterns across hippocampal neurons. We used the same LDA decoding approach as described above (time-resolved five-fold cross-validation, averaging firing rates across a sliding 500 ms window), collapsing across NAM HIT and AM+ trials and using firing patterns across a given participant's hippocampal neurons as features to discriminate buildings vs. landscapes. As shown in Fig. 5a, results revealed above-chance scene discrimination starting at 130 ms post stimulus onset ($P_{cluster} = .003$ vs. surrogates), with peak decodability of ~13% above chance at 440 ms. Applying the same decoding approach to distinguish associative vs. nonassociative retrieval processes (AM+ vs. NAM HITs, now collapsing across buildings and landscapes) showed that process-specific population codes started setting in at ~270 ms, with peak decodability of ~14% above chance at 820 ms ($P_{cluster} = .001$ vs. surrogates, Fig. 5b). Note that in addition to the main factor of interest, i.e. associative vs. nonassociative retrieval processes, AM+ trials and NAM HITs also differ with regard to the presence vs. absence of the grey square. Two considerations render it unlikely that this factor

would drive decodability though. First, the decoding performance peaked after 800 ms, at which point basic visual differences should only have marginal impact compared to earlier time points. Second, around the same time point (starting at 700 ms), hippocampal firing was significantly stronger for AM+ than AM− trials (Fig. 2c), with all visual input being identical between these two trial types. We thus suggest that the decodability shown in Fig. 5b reflects successful deployment of associative retrieval processes specific to AM+ trials. In any case, pinpointing the temporal dissociation of scene type and memory task decodability, a repeated-measures ANOVA with the factors Time Window (early $_{(0–1 sec)}$, late $_{(1–2 sec)}$) and Decodability (scene type, retrieval task) revealed a significant interaction ($F_{(1,14)} = 9.23$, $P = .009$) in the absence of a Time Window or Decodability main effect (both $F_{(1,14)} < = 2.01$, $P > = .178$). This interaction emphasises the relative increase in scene type decodability early in the trial and the relative increase in retrieval task decodability later in the trial. Note that this interaction was not contingent on the precise definition of early and late time windows (e.g., 0–.5 sec vs. 1–1.5 s: $F_{(1,14)} = 7.69$, $P = .015$; .25–.75 s vs. 1.25–1.75 s: $F_{(1,14)} = 9.32$, $P = .009$).

Analogous to the across-neuron correlation analysis performed for EC reinstatement (Fig. 3c), we next plotted the effect size for a given hippocampus neuron's task difference (AM+ vs. NAM HIT, independent samples t-test on firing rates averaged across the first 2 s of retrieval) against that neuron's scene type difference (buildings vs. landscapes, again independent samples t-test on firing rates averaged across the first 2 s of retrieval). As shown in Fig. 5c, there was no significant correlation between the two effects across neurons (Spearman r = .11, P = .105), suggesting that hippocampal scene-discrimination vs. associative retrieval effects are not only temporally dissociated, but are unlikely to be carried by the same neuron populations. This result is reminiscent of 'visually-selective' vs. 'memory-selective' neurons in human MTL recordings reported previously[24]. Interestingly, we observed no topographic effects in the anatomical distribution of scene-selective vs. memory-selective hippocampus neurons—in fact the most pronounced scene-selective neuron and the most pronounced memory-selective neuron in our sample were found on the same microwire bundle in the same participant, i.e. within a radius of a few mm at most (Fig. 5c).

**Discussion**
Returning to our opening example, our findings elucidate the mechanisms through which the human hippocampal-entorhinal cell circuitry enables successful memory retrieval in a scene cue

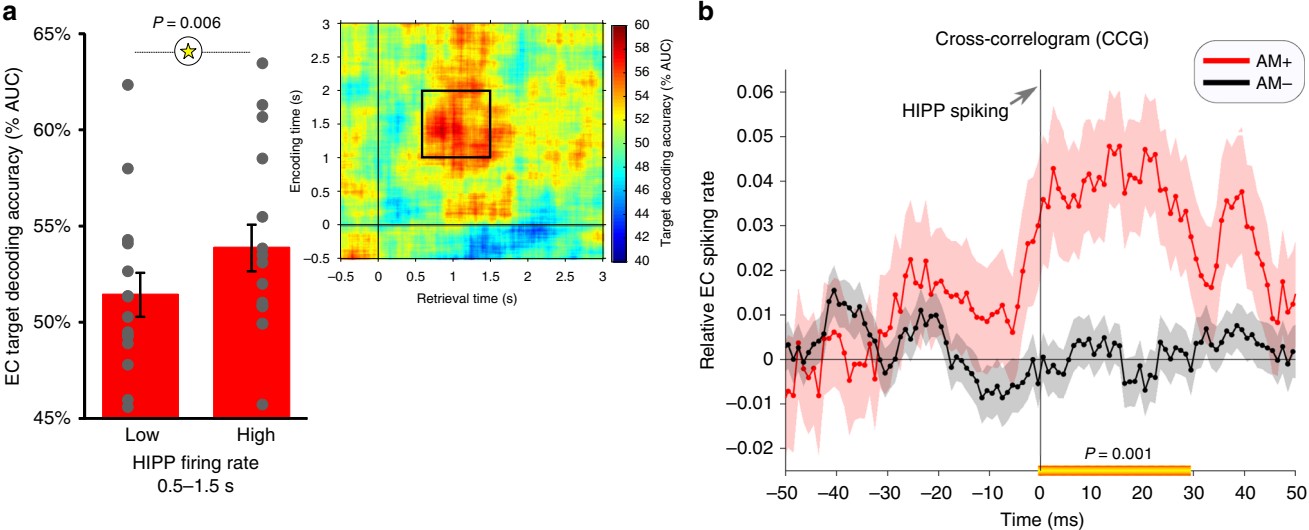

**Fig. 4** Relationship between HIPP and EC retrieval effects. **a** EC target reinstatement (inset) is greater for AM+ trials with higher HIPP firing rates. Bars represent mean ± SEM across participants, dots represent individual participants. **b** Cross-correlogram (CCG, shift-predictor-corrected), depicting the relative firing rate of EC neurons in a 100 ms interval around HIPP spiking during AM+ (red) and AM− (black) trials (mean ± SEM across within-hemisphere HIPP-EC neuron combinations, $n = 1798$). Note the significant increase of EC firing within 30 ms after HIPP firing, significantly exceeding both shift-predictor baseline (comparison vs. 0) and shift-predictor-corrected AM- trials (both $P_{cluster} = .001$, corrected for multiple comparisons across time)

('beach sunset')—object target ('cocktail') scenario: Upon arrival of visual information in the hippocampus, one population of neurons processes the content of the scene cue. Shortly thereafter, a separate process sets in, carried out by another population of hippocampal neurons and geared towards coordinating object reinstatement among entorhinal cortex neurons. Our findings thus track the evolution of human recollection with unprecedented precision and provide direct empirical evidence for a role of the hippocampus in orchestrating cortical pattern completion. Concurrently, our results reconcile two seemingly incompatible views on hippocampal function, emphasizing either perceptual/constructive scene processing[5–7] or associative/relational memory processing[8,10]—both functions appear to co-exist, but are carried out by different subpopulations and unfold at different time points in the course of memory retrieval.

Our first analysis compared hippocampal engagement for successful nonassociative memory (NAM, HITs and CRs) with successful associative memory (AM+). Hippocampal firing rates during recognition memory have been investigated previously[25–27], albeit without directly comparing associative vs. nonassociative tasks or examining hippocampal-entorhinal dynamics. As shown in Fig. 2b, hippocampal firing rates peaked at 500 ms for all three conditions, but critically were more sustained between 500 and 1500 ms for AM+ trials only. Within the same time window, we observed a relative increase in hippocampal firing rates for AM+ vs. AM− (Fig. 2c). The time course of the AM+ effect is consistent with the time course of recollection signals in scalp and intracranial Electroencephalography (EEG) studies[28] (for review, see ref. [29]). While we observed no reinstatement of the target object representation across hippocampal populations, hippocampal firing rates from 500 ms to 1500 ms predicted the strength of reinstatement among EC assemblies (Fig. 4a). This finding mirrors results from a previous fMRI study, which showed that hippocampal activation predicts the extent of reinstatement of a target scene image in parahippocampal cortex, without scene reinstatement in hippocampus itself[30]. Together, these results point to a role of the hippocampus in coordinating reinstatement/pattern completion

in cortical modules, which is further corroborated by the current finding that single hippocampal spikes temporally precede spikes in EC for AM+ but not for AM− (Fig. 4b).

Given that hippocampal neurons did not show object reinstatement, nor coded for object identity in this paradigm (Supplementary Figure 2), one interesting question is how the association between a scene cue and the target object is formed during encoding. In other words, how is the hippocampal index[9,12,13] that allows for subsequent pattern completion established in the first place? Computational models[31] and empirical data in animals[32] and humans[33] emphasize that the hippocampus codes information in a highly conjunctive manner, i.e. by assigning a unique code to new item-context combinations, even if the item itself is a repetition. As such, object 1 shown with scene A (trial n) would elicit a different assembly pattern than object 1 shown with scene B (trial $n + 1$). That is, while hippocampal populations might show relatively poor decodability for general object identity, they may well code for a given trial identity, in the sense of a true episodic index. A recent study showed evidence for this notion via hippocampal field recordings, demonstrating that the pattern similarity between retrieval of a particular word-colour combination and its encoding counterpart was greater than the similarity with any other encoding trial featuring the same colour[28]. Nevertheless, specific semantic representations that are context-independent have likewise been found at the level of single hippocampal neurons[34], consistent with the present finding that hippocampal population codes allow decoding of two different scene categories (buildings vs. landscapes, Fig. 5a) across trials. One interesting question for future research is whether firing patterns in 'visually-selective' neurons tend to generalise across episodes, whereas 'memory-selective' neurons, preferentially engaged in associative memory processes, provide trial-unique episodic indices.

The fact that hippocampal populations did not code for object identity might at first seem at odds with previous single-unit studies in humans, showing selective hippocampal responses to processing[35–38] and recall[39] of particular object stimuli. However, a comparison of stimulus category preferences across

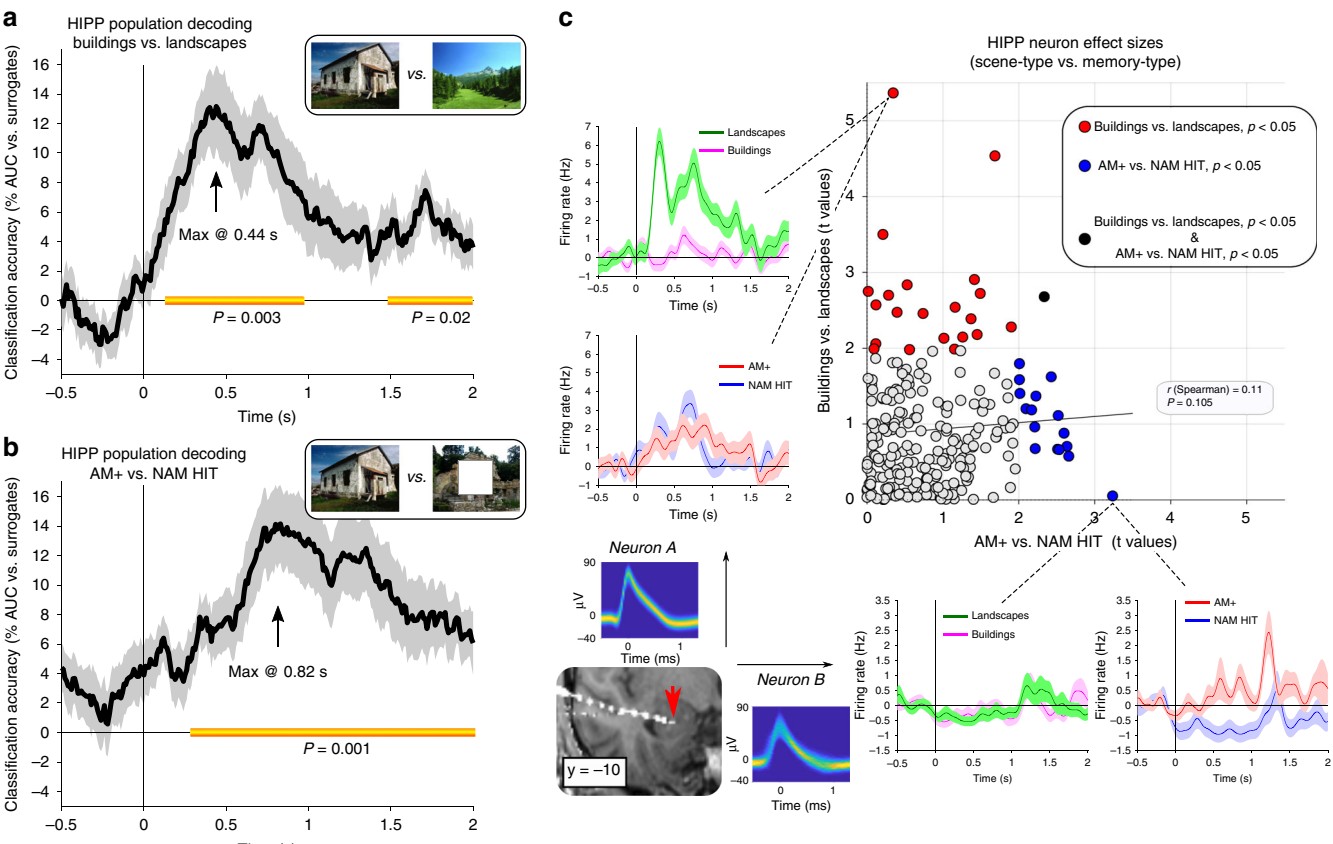

**Fig. 5** Two independent populations of hippocampal neurons code for scene type and memory type. **a** Decoding of scene type (buildings vs. landscapes, both collapsed across NAM HIT and AM+) across time (10 ms steps; x-axis reflects decoding time points after averaging firing rates in a +/−250 ms window). **b** Decoding of memory type (AM+ vs. NAM HIT, both collapsed across buildings and landscapes). **c** Scatter plot shows each HIPP neuron's effect size (abs t-values) for the contrast of AM+ vs. NAM HIT (x-axis) and the contrast of buildings vs. landscapes (y-axis), based on baseline-corrected mean firing rates from 0–2 s after stimulus onset. Black line is the least-squares regression fit. Filled circles denote neurons showing a scene-type effect (red), a memory-type effect (blue) or both (black) (calculated via independent samples t-tests across trials for each neuron, P < .05 uncorrected). Insets show mean instantaneous firing rates for the two neurons showing the strongest effects, both incidentally stemming from the same participant and the same microwire bundle (red arrow; shown on the normalized post-implantation MRI). Waveforms of the two neurons' respective action potentials are depicted as temperature-scaled density plots. Scene images obtained from https://commons.wikimedia.org under the CC-BY-SA-3.0 license

mediotemporal regions showed that objects are underrepresented in the hippocampus compared to EC[40]. Besides, these studies typically quantified how many neurons in a given region distinctively responded to one or more stimuli from a large stimulus set, which is not what we did in the current study. Instead, we here embraced a neuronal population approach rather than a single neuron approach[41–43], examining—participant by participant—whether the population code across hippocampal or EC neurons can distinguish between two object exemplars (Fig. 3, S2), scene categories (Fig. 5a) or memory tasks (Fig. 5b).

Our EC data provide the first evidence for memory-guided episodic reinstatement across neuronal populations in this region. Interestingly, while univariate results showed that EC firing rates distinguished both between AM+ vs. NAM HITs and between NAM HITs vs. CRs (similar to patterns reported for perirhinal cortex[44–46]), there was no difference in EC firing rates between AM+ and AM− trials (Supplementary Figure 1). We speculate that this could be due to the fact that (i) only two target objects were used in each participant and (ii) the majority of AM- trials consisted of incorrect object responses rather than '?' responses (see behavioural results). That is, even in the case of AM- trials, EC neurons are recruited to represent the non-target object.

Indeed, decoding accuracies for AM− were not only significantly lower than for AM+ (Fig. 3b), but were partially negative (Supplementary Figure 3b), hinting towards reinstatement of the non-target object. Finally, the cross-correlogram (CCG) results suggest that despite comparable overall firing rates, EC spikes during AM− trials are less tightly locked to preceding hippocampal spikes than during AM+ trials (Fig. 4b).

Given its functional-anatomical proximity to the hippocampus, EC has been postulated as the first site of target reinstatement[13]. That said, it is likely for successful recall to encompass reinstatement across multiple cortical modules as recollection unfolds. Indeed, a recent study recording from neurons in the human anterior temporal lobe has shown evidence for reinstatement of semantic information in this region during successful recall[47]. Simultaneous recordings from lower-level perceptual and higher-level semantic regions would allow tracking the gradual emergence of a full-blown memory, alongside a possible reversal of information flow from encoding to retrieval[48].

Lastly, an interesting question for future research is how the dynamics between hippocampus and EC would manifest had we reversed the role of objects and scenes in our paradigm. In other words, if trial-unique objects were used to cue one of two target

scenes, how would this affect the roles of the two regions in successful recall? For the current scenario, our data suggest that one set of hippocampal neurons represents the scene cue information before a second set of hippocampal neurons is deployed to coordinate reinstatement of the target object in EC. We thus speculate that if objects served as cues, we would first see a response in object-tuned EC assemblies. This cue-specific EC response would then be followed by deployment of the same set of process-specific ('memory-selective') hippocampal neurons observed here (Fig. 5b), which would in turn coordinate reinstatement of target scene representations among scene-specific ('visually-selective') hippocampal neurons (alongside scene-specific cortical modules[49]). In other words, we would postulate a domain-general set of hippocampal neurons dedicated to coordinating associative retrieval/pattern completion[50], but the time point of engagement of EC neurons vs. scene-selective hippocampal neurons could reverse as a function of the assignment of objects and scenes to cues and targets[51].

## Methods

**Participants**. Sixteen participants (7 male; 22–54 years old) undergoing treatment for pharmacologically intractable epilepsy were implanted with chronic depth electrodes to localize the seizure onset zone for possible surgical resection. All studies conformed to the guidelines of the Medical Institutional Review Board at the University of Bonn. Informed written consent was obtained from each participant. Electrode locations were planned exclusively based on clinical considerations and included the anterior half of the hippocampus (i.e., hippocampal head and body) as well as entorhinal cortex.

**Paradigm**. A schematic of the experiment is shown in Fig. 1. Each experimental session consisted of 12 runs (4 nonassociative memory (NAM), 8 associative memory (AM)) containing an encoding block, a delay block and a retrieval block. NAM and AM runs were arranged such that one NAM run alternated with two consecutive AM runs, with the assignment of the first run (NAM or AM) counterbalanced across participants. All trials began with a .5 s fixation cross, followed by a 2 s (NAM) or 3 s (AM) stimulus display, followed by a 3-choice response screen (left arrow, down arrow, right arrow; self-paced). During retrieval trials, a .5 s feedback screen followed the button press in which a green fixation cross was shown for correct responses, a red fixation cross for incorrect responses and a white fixation cross for "don't know" responses. Before starting the next trial, a blank-screen inter-trial interval with a jitter of .1-.4 s was presented. The study-test delay consisted of a 30 s interval in which participants were instructed to relax while the remaining rest time was shown on the screen.

In NAM runs, participants saw 20 scene images (depicting buildings or landscapes) during encoding and indicated whether they would like to visit the shown environment (left arrow), would not like to visit the environment (right arrow) or were not sure (down arrow). At retrieval, participants saw the same 20 scene images randomly intermixed with 20 novel scene images and indicated whether they thought the image was old or new, yielding the memory outcomes HITs (old images correctly identified as old), MISSes (old images incorrectly identified as new), Correct Rejections (CRs, new images correctly identified as new), False Alarms (FAs, new images incorrectly identified as old) and "don't know" responses. In AM runs, participants saw 10 new scene images (from the same buildings/landscapes set) during encoding. Superimposed on the scene was one of two object images, surrounded by a grey square. The choice of object images was customized for each experimental session based on a previous screening session to identify response-eliciting stimuli in any of the recorded neurons (see below). For each encoding trial, participants indicated whether they thought the given object was (left arrow) or was not (right arrow) likely to be encountered in the given environment. In case they could not decide, they were allowed to indicate "not sure" (down arrow). At retrieval, participants saw the same 10 scene images (in random order) with the grey squares superimposed, but critically the objects themselves were removed. After a 3 s retrieval period, participants saw the verbal labels of the two objects and indicated which of the two they thought had been paired with the given scene during encoding, yielding the memory outcomes correct Associative Memory (AM+) and incorrect as well as "don't know" responses (the latter two referred to as AM−). In total, participants were presented with 80 old and 80 new scene images during nonassociative retrieval and 80 old scene images during associative retrieval.

Regarding the choice of object stimuli, the screening sessions were usually conducted first thing in the morning and would then be used for different aspects of the various subsequent experiments conducted that day. For the current experiment, we chose two object images (out of 100 presented during a screening session) that elicited a strong response in any of the recording contacts, irrespective of the anatomical location of the responding contact. In that sense, there was no a priori bias to favour object stimuli preferred by EC or hippocampus. Nevertheless,

we also quantified post-hoc how many EC neurons and how many hippocampal neurons showed stronger firing rates for one of the two object images over the other during the encoding portion of this experiment. Specifically, we averaged the firing rates for the entire 3 s of an encoding trial and compared, neuron by neuron, the firing rates for the 40 encoding trials showing object 1 with the 40 encoding trials showing object 2 (independent samples t-test, $P < .05$, two-tailed). Results showed that an average of 1.2 hippocampus neurons per participant (SEM = 0.6) and 2.2 EC neurons per participant (SEM = 0.5) showed a statistically significant response increase for one object over the other. While this difference only showed a statistical trend ($t_{(15)} = 1.78$, $P = .096$), this univariate effect would still impact multivariate decodability, as there are on average more 'diagnostic' units in EC than hippocampus. We thus refrained from directly comparing object decodability between the two regions.

**Electrophysiological recordings**. Recordings were obtained from bundles of nine microwires each (eight high-impedance recording electrodes, one low-impedance reference, AdTech, Racine, WI) protruding ~4 mm from the end of each depth electrode. The differential signal from the microwires was amplified using a 256-channel Neuralynx ATLAS system (Bozeman, MT), filtered between 0.1 and 9000 Hz, and sampled at 32 kHz. These recordings were stored digitally for further analysis. Spike detection and sorting was performed after band-pass filtering the signals between 300 and 3000 Hz[37,52]. Sorted units were manually confirmed and classified as single units (SU), multi-units, or artifacts based on spike shape and variance, peak-amplitude-to-noise level, the inter-spike interval (ISI) distribution of each cluster and presence of a refractory period. Depending on which implantation sites were actually chosen by the responsible clinicians, the number of microwires analyzed per patient ranged between 24 and 48. Anatomical localization of microwires was determined based on the post-implantation CT scan co-registered to the pre-implantation MRI scan, both normalized to MNI space with SPM12. The end of the corresponding depth electrode was visually identified and a 3-mm-radius sphere was placed 4 mm medial to the electrode tip. For visualization in Figs 2a and 3a, participants' MRIs and EC/hippocampus spheres were averaged. Percent coverage refers to the number of participants with EC/hippocampus spheres at a given voxel. Note that for sagittal slices, left hemisphere spheres were projected onto the right hemisphere.

**Analysis**. For continuous representations of instantaneous neuronal firing, spike trains were convolved with a Gaussian kernel (50 ms width, 1 ms time resolution). Resulting time series were baseline-corrected by subtracting firing rates in the interval from −.5 to 0 s and subjected to parametric tests at each time point. Correction for multiple comparisons across time (stimulus onset until end of trial) was performed using FieldTrip's nonparametric cluster-based permutation method (1000 randomisations)[19]. Pairwise comparisons via t-tests were two-tailed.

Mulitvariate pattern analysis was performed via linear discriminant analysis (LDA), using—at each individual time point—the firing rates of all neurons in EC or hippocampus as features (after averaging across a sliding 500 ms window and z-scoring across trials instead of baseline normalization). For reinstatement analysis (Fig. 3b), a classifier was first trained on all 80 AM encoding trials and then applied to each AM+ retrieval trial, assessing the decoding accuracy during successful retrieval based on object-specific encoding patterns. This was done in an encoding time (training) x retrieval time (testing) fashion. For stimulus decoding at encoding or retrieval per se, k-fold cross-validation was performed. K was set to 5, and five repetitions were included to provide stability across different partitions of training and testing trials. For all decoding analyses, the outcome measure was % area under the curve (AUC) of a decoding Receiver Operating Characteristic (ROC)[53]. To enter decoding analyses, we required a minimum of three neurons per participant, resulting in exclusion of one participant for hippocampal decoding analyses.

For cross-correlogram (CCG) analyses, we obtained spike occurrence histograms for EC neurons with respect to individual hippocampus neuron spikes (±50 ms, 1 ms bin size), using all within-hemisphere combinations of EC/hippocampus neuron pairs in a given participant (resulting in a total of 1798 combinations/histograms). Each participant provided an average of 112 (SEM = 35) EC/hippocampus combinations, with an average of 12 (SEM = 2) EC neurons and an average of 12 (SEM = 3) hippocampal neurons contributing. Prior to entering statistical analysis, histograms were smoothed with a 10 ms running average. To control for trial-invariant firing properties across neurons, a 'shift-predictor' baseline CCG was derived by scrambling trial orders such that seed spikes came from trial n and target spikes from trial $n \pm 1$. The shift-predictor CCG for each neuron combination was then subtracted from its real CCGs and the resulting shift-predictor-corrected CCG was tested against zero across-neuron combinations.

## Data availability

The data that support the findings of this study are available from the corresponding author upon reasonable request.

## Code availability

Custom computer code used to generate the results of this study is available from the corresponding author upon reasonable request.

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

## Acknowledgements

This research was supported by a Wellcome Trust/Royal Society Sir Henry Dale Fellowship to B.P.S. (107672/Z/15/Z), Volkswagen Foundation and German Research Council grants to F.M. (MO930/4-1, SFB1089), and a Swiss National Science Foundation grant to T.P.R. (P300P1_161178). We thank B. Samimizad for help with spike-sorting.

## Author contributions

B.P.S. and F.M. designed research. J.N., T.P.R. and F.M. collected data. B.P.S. and T.P.R. analysed the data. B.P.S., T.P.R. and F.M. wrote the manuscript. C.E.E. recruited patients. J.B. and F.M. performed neurosurgical procedures.

## Additional information

**Competing interests:** The authors declare no competing interests.

