## [Peer Review File · Nature Communications]

Reviewers' comments:

Reviewer #1 (Remarks to the Author):

Review of Staresina et al

The authors present recordings from the human hippocampus and entorhinal cortex investigating the formation and recall of associative and non-associative memories. In the non-associative memory task the patients viewed scenes, then were tested with old and new scenes in a subsequent recall test. In the associative memory task, they viewed scenes with a superimposed object. During recall they were shown the scene without the object and had to recall the associated object. The results indicate that hippocampal firing was increased during the associative memory task compared to the non-associative task. During the associative task, the identity of the associated object could be decoded from the firing-rates of cells in the entorhinal cortex, and the authors suggest that the firing of entorhinal neurons is driven by hippocampal firing. Furthermore, hippocampal firing contained information about the type of scene and they showed that information about scene type and memory type (or task) were carried by largely independent groups of neurons in the hippocampus. The results are based on a large data-set of neurons from many patients and are based on sound statistical analyses. The paper is also very clearly written and is easy to follow. The authors interpret the data as evidence that hippocampal cells responding to the scene context drive associated memory responses in the entorhinal cortex, but that conclusion is not fully supported by the data (see below). Yet, this is a wonderful study and the result is of great importance for our understanding of how human associative memories are formed and recalled. I have a few points of clarification and there is a mistake in the interpretation of cross-correlation functions, which could be easily corrected.

1) As I understand it only two objects were used during the AM encoding task, these objects were chosen during a pre-screening session, the data of which are not shown. How exactly were these objects chosen – in particular did the objects drive HC or EC activity during the screening session? If objects were chosen that drove tuned responses in HC cells then it seems strange that the identity of the object could not be decoded from the HC during the AM encoding task (Fig. S2). Vice versa, if scenes were specifically chosen to match the tuning of HC cells and object to match the tuning of EC cells, some of the results (e.g. Fig. S2) are the direct results of the design of the study, not necessarily because EC and HC have different functions.

- If this is true, the reader should be informed from the start that that authors pre-selected HC cells tuned for the scenes and EC cells tuned for the objects.

- If the objects were indeed chosen because of tuned activity in a few 'concept cells' in the HC or EC, did these cells show any differential activity during AM+/- trials similar to the single example shown in Figure 4?

- Given that HC doesn't seem to encode object identity here – by which mechanism is the association formed between the picture and the scene during encoding? In the authors final interpretation, a separate group of cells in the HC are suggested to be responsible for driving the object specific effect in the entorhinal cortex, but how do the authors envisage this happening if the HC cells do not appear to encode the object identity?

2) Related to this: In the introduction, the authors suggest that the HC are indexes to memories in EC. This is supported by the data. However, if the HC neurons are selected because they are selective for scenes and EC neurons for the objects, then the design is biased for detecting that HC neurons drive EC neurons. Imagine the opposite task: the subject sees one of the objects and has to recall the scene. Now EC might actually need to drive the HC cells. In that case, the EC cell would provide indices to memories in HC. The authors should acknowledge that their design was therefore biased to support the "hippocampal index hypothesis".

3) The authors suggest that HC cells drive EC cells (e.g. in the abstract). However, such a conclusion of causality is not supported by the data. This would require some kind of intervention with the neuronal activity.

- The same mistake at the top of p. 6 "our findings reveal that neurons in the human hippocampus exert a driving influence"

- In the legend to Fig. 3 the authors state that the HIPP cell shows sustained firing during AM+ at

~750 ms. Is that the data in Fig. 3b? It was not entirely clear to me whether this data came from the encoding or decoding epoch.

4) There seems to be a difference in difficulty between the two tasks with the associative task being a bit more difficult. It seems not so likely to me, but the authors should discuss the role of this possible confound.

- Was there a RT difference between the two tasks?

5) The authors make a mistake in the interpretation of the cross-correlation function when they state that subtraction of the shift-predictor rules out a common driver. That is wrong. The shift predictor does not rule out a common driver but it can only control for common effects on two neurons caused by the onset of a stimulus or repetitive task events.

- Why did the authors only use trials $n \pm 1$ to compute the shift predictor? Most studies simply take all trial combinations. Does that make a difference in the analysis? If so, it could be related to non-stationarities in the signal.

6) In the example neuron in Fig. 3b there is higher activity in the HIPP cell during AM+ than NAM encoding. Was that generally the case?

- Were there general differences in HIPP and EC during encoding between AM+ and AM- trials? I.e. did higher activity predict better encoding?

- In Figure S4 there is an intriguing effect of an overall elevation of the correlation function in the AM+ trials during encoding (the left plot). This presumably means that the firing rates were better correlated across trials, without fine temporal structure (see e.g. Bair et al. J. Neurosci 2001 for an excellent explanation of how this works). However, whether this effect can be observed in the analysis depends on the details on the cross-correlation analysis (different labs make different choices regarding normalization and subtraction of the baseline), details which should be added to the Ms. Do the authors indeed observed such an increase in the general across-trial correlation on AM+ trials?

7) In Fig. 4 there are differences between the visual stimuli (presence of an occluder) that could explain the difference between AM and NAM activity. This is not so likely given the timing of the effect on firing rate, but it should nevertheless be briefly discussed.

- In the AM trials, the authors used 3s presentation and in the NAM trials 2s. Is this a confound? If not, please explain why.

8) The patients make a reasonable number of errors during the AM task, do the EC classifiers predict the incorrect object during these error trials?

9) What happens to HIPP and EC activity during the 30s delay? Do the neurons code for the last presented item?

10) In every trial, there were two encoding epochs and two decoding epochs. I suppose that the data were collapsed across epochs for the analysis. Were the effects also visible when the first and the second epochs were evaluated separately? Were there any differences between the epochs?

11) The authors generally used a decoding approach. How good were these effects visible when the tuning of individual neurons was used instead?

Small points:

- Fig. 2e: explain the meaning of the black and grey bars

- Fig. 4c: can you make the scale on the x-axis and y-axis similar? That would provide a more realistic impression of the degree of tuning for the two factors

- Analysis: explain the meaning of "baseline corrected". Corrected how?

Reviewer #2 (Remarks to the Author):

The paper by Staresina et al uses single unit recording data from epileptic patients to look at the dynamics of neural activity in the hippocampus and entorhinal cortex during associative memory recall, and compares the activity with a simpler recognition task. This sort of research remains rare, because only a relatively small number of research centers are able to do this sort of investigation. But here, the authors add a number of twists that mean that the results are particularly intriguing and novel.

Essentially, they describe prolonged activity in the hippocampus on trials where the patient is required to remember whether a particular object was associated with a particular scene. Furthermore, this increased hippocampal activity is associated with selective activation of the entorhinal cortex (EC) – and thus may be perhaps the first clear demonstration of the action of a hippocampal – EC circuits in episodic memory. The study is particularly unusual in using cross correlations between pairs of hippocampal and EC neurons to provide evidence for linkage between the populations of neurons.

Given the novelty of the results and their potential implications for the understanding of the neural mechanisms underlying episodic memory, I am essentially quite enthusiastic about this paper, which is also well written.

I nevertheless have some questions.

1) The experiments rely on having neurons that have already been demonstrated to have selectivity for particular objects – for example, scorpion vs raspberries. However, there is no description of how this selectivity was demonstrated. In particular, when they pair a particular object with an image, and then test with a pair of names (“scorpion” vs “raspberries”), is it the case that there are always neurons that are individually selective to both the test items? If the patient had been previously tested with lots of images of scorpions, but not with raspberries, this would unbalance the protocol. Can the authors confirm that all the test images are equally familiar?

2) In the legend to figure 2b, it is stated that it plots “Mean hippocampal firing rates”, but since the firing rate before the stimulus comes on is zero, I presume that the firing rates have been baseline corrected. The responses look quite neat, but when you see that we are talking about a maximal increase of 0.7 Hz across the population, this seems to be fairly weak. I presume that some individual neurons must have substantially larger increases in firing rate to achieve this. Some additional information about the reliability of such responses would be appreciated.

3) In the methods, the authors say that the cross-correlelogram data was obtained using all within-hemisphere combinations of HIPPO/EC neuron pairs in a given participant (resulting in a total of 1798 combinations/histogram). Could the authors say how many hippocampal and EC neurons were involved?

Reviewer #3 (Remarks to the Author):

In this study, Staresita et al report on simultaneous single-neuron recordings performed in the human hippocampus and entorhinal cortex (EC) during two types of memory retrieval tasks. They report that i) hippocampal cells responded more during successful recall compared to successful recognition, ii) the identity of the recalled item could be decoded from EC, and iv) cross-correlation analysis indicates that hippocampal activity precedes that in EC during successful recall. The scientific novelty of this paper is principally the EC results.

The work is based on rarely available high-quality data and the experiments are performed well. The key novel technical aspects of this study are comparison of the same neurons in two tasks (recognition, recall) and the use of simultaneous EC-Hippo pairs of cells to study recall. This result is potentially of wide interest to the community, given the unique ability to study memory recall at single-neuron level in humans.

The results are interesting and the data of high quality. However, I am concerned about the robustness of the results and the way they are presented that limit my enthusiasm. The principle results on the EC (i.e. Fig 2d) are of small size and significance and not supported by single-neuron analysis or correlations with behavior. In contrast, the hippocampal results appear

substantially more robust (Fig 4). It is thus unclear how robust the reported effects in EC are and whether they are of relevance for memory. Also, the writing is unnecessarily strong and does not properly relate these findings to earlier work in several instances. In conclusion, while interesting, I do not find that there is sufficiently robust support for the strong claims made in this study to warrant publication in this form. Having said that, an extensively revised version might be able to overcome these concerns.

Major issues:

1. Robustness. Decoding accuracies are small (AUC values ~ 0.52). This in particular applies to Fig 2d, which is the key result. The impression of marginal significance is strengthened by an absence of exact p-values for the significant clusters. Overall it is not clear whether the statistical approach taken was appropriate.

2. Absence of single-unit analysis for EC neurons and absence of comparable analysis for EC vs Hippo. Apart from one example (Fig 3), the principle effects here are not reported for individual neurons. The results on hippocampal neurons in Fig 2b are averaged across all neurons. That seems like an odd choice, given that there are a variety of types of hippocampal neurons (as can be seen in Fig 4). Something like Fig 4 is needed for EC. In particular, how many EC neurons show evidence of reactivation due to free recall? Do any hippocampal neurons show such a signal? Also, some analysis is shown for one area (Hippo, Fig 2b) but not the other (is this also the case for EC?). This limits ability to compare results between areas.

3. Lack of citation of prior work and unfounded claims. Aspects of the work here have been shown before, in particular recollection-triggered activation of selective cells in entorhinal cortex and hippocampus (Gelbard-Sagiv et al. 2008 Science), selectivity of MTL cells to categories and objects (Fried 1997, Kreiman et al. 2000, Nat Neurosci, Quiroga et al 2005 Nature), and selectivity of MTL cells to familiarity/recollection. Also, there is work on simultaneous EC-hippo recordings in rodents addressing issues related to reinstatement. None of these are cited/discussed. A second writing issue is that unwarranted statements of causality are made, none of which can be substantiated without causal intervention. For example: "our findings reveal that neurons in the human hippocampus exert a driving influence on EC target sites". In general, the scholarship of this paper needs to be improved.

4. Potential confound for cross-correlation analysis between simultaneously recorded EC-Hippo cell pairs. I am uncertain whether this analysis does not suffer from the following confound. As Fig 3 shows, neurons in hippocampus respond with a latency earlier than that in EC. Thus, there is a common input, but with a latency offset. It is unclear whether this is what drives the directional effects seen in the cross-correlation analysis (rather than a true spike-by-spike directional influence).

5. Object identity encoding in EC. Object identity could be decoded from Hippocampus, but not EC. But this is not what the prior literature shows? This statement is based on what is shown in Fig S2. But this figure is far from convincing due to marginal differences. How many neurons encode objects in each area? Also it is at odds with the literature, which shows object identity coding in both EC and hippocampus (i.e. Mormann et al 2008 J Neurosci).

6. Was the extent of reinstatement in EC indicative of successful recall? More extensive support for the very interesting and novel result of a potential role of reinstatement in EC for memory recall would strengthen the paper.

Minor:

1. In some plots the units are confusing. For example, in Figs 3-4, there are negative firing rates?
2. All statistical claims need to be accompanied by an effect size and p-value
3. Why did the decoder in Fig 2d achieve what appears to be significant below-chance performance shortly after stim onset?

Responses to Reviewers

Reviewer #1 (Remarks to the Author):

Review of Staresina et al

The authors present recordings from the human hippocampus and entorhinal cortex investigating the formation and recall of associative and non-associative memories. In the non-associative memory task the patients viewed scenes, then were tested with old and new scenes in a subsequent recall test. In the associative memory task, they viewed scenes with a superimposed object. During recall they were shown the scene without the object and had to recall the associated object. The results indicate that hippocampal firing was increased during the associative memory task compared to the non-associative task. During the associative task, the identity of the associated object could be decoded from the firing-rates of cells in the entorhinal cortex, and the authors suggest that the firing of entorhinal neurons is driven by hippocampal firing. Furthermore, hippocampal firing contained information about the type of scene and they showed that information about scene type and memory type (or task) were carried by largely independent groups of neurons in the hippocampus.

The results are based on a large data-set of neurons from many patients and are based on sound statistical analyses. The paper is also very clearly written and is easy to follow. The authors interpret the data as evidence that hippocampal cells responding to the scene context drive associated memory responses in the entorhinal cortex, but that conclusion is not fully supported by the data (see below). Yet, this is a wonderful study and the result is of great importance for our understanding of how human associative memories are formed and recalled. I have a few points of clarification and there is a mistake in the interpretation of cross-correlation functions, which could be easily corrected.

We thank Reviewer 1 for their endorsement of our manuscript and the very constructive comments.

1) As I understand it only two objects were used during the AM encoding task, these objects were chosen during a pre-screening session, the data of which are not shown. How exactly were these objects chosen – in particular did the objects drive HC or EC activity during the screening session? If objects were chosen that drove tuned responses in HC cells then it seems strange that the identity of the object could not be decoded from the HC during the AM encoding task (Fig. S2). Vice versa, if scenes were specifically chosen to match the tuning of HC cells and object to match the tuning of EC cells, some of the results (e.g. Fig. S2) are the direct results of the design of the study, not necessarily because EC and HC have different functions.

- If this is true, the reader should be informed from the start that that authors pre-selected HC cells tuned for the scenes and EC cells tuned for the objects.

We thank Reviewer 1 for raising this important point. Critically, the screening session did not include any scene images, so all results pertaining to scene coding in the hippocampus (Figure 4, now Figure 5) are entirely unbiased by stimulus selection. With regard to object stimuli, the rationale of the

screening session was as follows: These sessions were usually conducted first thing in the morning and would then be used for different aspects of the various subsequent experiments conducted that day. For the current experiment, we chose 2 object images that elicited a strong response in any of the recording contacts, irrespective of the anatomical location of the responding contact. In that sense, there was no a priori bias to favour object stimuli preferred by EC. What's more, given that the screening session and the main experiment were often conducted several hours apart, it is not certain that a neuron that responds to a particular object during the screening session still responds to that object during the main experiment (Niediek et al., PLoS one 2016). Lastly, the definition of 'response-eliciting' is somewhat arbitrary and highly dependent on statistical criteria. That is, clear-cut stimulus selectivity as shown for the single EC unit in Figure 3 (selectively responding to raspberries and not scorpion) are the exception, arguably due to the relatively sparse sampling afforded by human single cell recordings.

That said, we now quantified how many EC units and how many HIPP units showed stronger firing rates for one of the two object images over the other. Specifically, we averaged the firing rates for the entire 3 sec of an encoding trial and compared, unit by unit, the firing rates for the 40 encoding trials showing object A with the 40 encoding trials showing object B (independent samples t test, $p < .05$, two-tailed). Results showed that an average of 1.2 HIPP units per participant (SEM = 0.6) and 2.2 EC units per participant (SEM = 0.5) showed statistically different response profiles for one object over the other. While this difference only shows a statistical trend ($t(15) = 1.78$, $P = .096$), this univariate effect will still impact multivariate decodability as there are on average more diagnostic cells in EC than HIPP. This is now detailed in the Methods section (page 13-14).

Consequently, we now refrain from directly comparing decodability of objects in EC vs. HIPP. Nevertheless, the fact that HIPP cells performed at chance level for object decoding in our paradigm is still interesting in light of the last question raised by the Reviewer (how can HIPP cells drive recall if not coding for object identity; see below), so we still report the lack of decodability in the supplement (emphasizing though that this might be a consequence of the selected stimuli). Also note that the reinstatement effects in EC (main Figure 2d, now Figure 3) are not biased in any fashion, as the key finding here is that EC response patterns from perceptual encoding return upon recall in the absence of visual object information.

- If the objects were indeed chosen because of tuned activity in a few 'concept cells' in the HC or EC, did these cells show any differential activity during AM+/- trials similar to the single example shown in Figure 4?

Given that the screening was not designed to specifically reveal concept cells in EC, clear-cut examples like the one shown in Figure 4 (now Figure 3d) are relatively rare in this study. In general, we think it is a particular strength of our study that we capitalise on regional population patterns rather than single cell effects (cf. Yuste, Nature Reviews Neuroscience 2015; Wallis, Trends in Cognitive Sciences 2018; both now cited). Nevertheless, we added a new analysis in which we plot each EC cell's effect size for object 1 vs. object 2 at encoding against the corresponding effect size during AM+ retrieval. Results of this analysis are quite compelling: First, across the 3-second retrieval period, there is indeed a positive correlation of effect sizes with the corresponding encoding effects ($r = .19$, $P = .006$). Importantly, no such correlation was seen for AM- trials ($r = .02$, $P = .787$). We now added this analysis to the main manuscript (Figure 3c) as well as to the supplement (Figure S3b).

- Given that HC doesn't seem to encode object identity here – by which mechanism is the association formed between the picture and the scene during encoding? In the authors final interpretation, a separate group of cells in the HC are suggested to be responsible for driving the object specific effect in the entorhinal cortex, but how do the authors envisage this happening if the HC cells do not appear to encode the object identity?

That's a very good question that requires a little speculation. First, as mentioned above, there may well be object-tuned concept cells in the hippocampus (as suggested by previous work), with our sampling merely having a relatively low yield. However, computational models (O'Reilly and Rudy, Psychological Review 2001) and empirical data in animals (Komorowski et al., Journal of Neuroscience 2009) and humans (Goh et al., Journal of Neuroscience 2004) also suggest that the hippocampus codes information in a highly conjunctive manner, i.e., assigning a unique code to new item-context combinations, even if the item itself is a repetition. As such, object 1 shown with scene A (trial n) would elicit a different response than object 1 shown with scene B (trial n+1). In other words, while HC cells might not encode the object identity, they may well encode the *trial* identity in the sense of a true episodic index. We now pick up this idea in a dedicated paragraph in the Discussion section (page 11).

2) Related to this: In the introduction, the authors suggest that the HC are indexes to memories in EC. This is supported by the data. However, if the HC neurons are selected because they are selective for scenes and EC neurons for the objects, then the design is biased for detecting that HC neurons drive EC neurons. Imagine the opposite task: the subject sees one of the objects and has to recall the scene. Now EC might actually need to drive the HC cells. In that case, the EC cell would provide indices to memories in HC. The authors should acknowledge that their design was therefore biased to support the “hippocampal index hypothesis”.

Because there was no selection of HC cells based on response profiles and no selection of scene stimuli based on HC responses, there is no bias for the effects we observed in HC. That said, we think the question of what would happen if we flipped the assignment of cue and target, i.e., use trial-unique objects as cues and two scene categories as targets, is very intriguing. We now raise this question in the Discussion section and speculate about possible scenarios (page 12).

3) The authors suggest that HC cells drive EC cells (e.g. in the abstract). However, such a conclusion of causality is not supported by the data. This would require some kind of intervention with the neuronal activity.

- The same mistake at the top of p. 6 “our findings reveal that neurons in the human hippocampus exert a driving influence”

The Reviewer is of course correct and we apologize for inferring a driving effect in the absence of intervention. The wording has now been adapted throughout the manuscript to only refer to temporal precedence and in the Results (page 8), we now explicitly state, “While this set of results is consistent with a role of hippocampus coordinating pattern completion in EC, we note that evidence for causality would require an experimental intervention approach.”

- In the legend to Fig. 3 the authors state that the HIPP cell shows sustained firing during AM+ at ~750 ms. Is that the data in Fig. 3b? It was not entirely clear to me whether this data came from the encoding or decoding epoch.

Yes, this referred to the example in panel b. However, we now restructured the figures to dedicate one Figure to the HIPP results (Figure 2) and another to EC results (Figure 3), obviating the need to highlight the exact timing of the single neuron example.

4) There seems to be a difference in difficulty between the two tasks with the associative task being a bit more difficult. It seems not so likely to me, but the authors should discuss the role of this possible confound.

- Was there a RT difference between the two tasks?

Given that we had participants withhold their response until prompted at the end of each trial, the RTs themselves are probably of limited informational value. Nevertheless, to address this question, we conducted a repeated-measures ANOVA with the factors Task (NAM, AM) and Accuracy (correct, incorrect – using HITs and MISSes for the NAM task) on RTs. Note that one participant was excluded from this analysis because they had no MISSes. Results showed a main effect of Accuracy ($F_{(1,14)} = 11.95$, $P = .004$; slower RTs for incorrect responses), but no main effect of Task ($F_{(1,14)} = 2.00$, $P = .179$) nor a Task x Accuracy interaction ($F_{(1,14)} = 0.010$, $P = .920$).

That said, as reported in the main manuscript, accuracy for AM was lower than accuracy for NAM, so Reviewer 1's concern about different task difficulties would still hold. However, if HC firing rates were merely driven by difficulty, one might expect a further increase for AM- trials, which are arguably more difficult than AM+ trials. Importantly though and as now shown in Figure 2c, there was a significant decrease for AM- relative to AM+, consistent with a 'retrieval success' function of HC. We now mention the potential caveat of task difficulty in the Results section (page 4).

5) The authors make a mistake in the interpretation of the cross-correlation function when they state that subtraction of the shift-predictor rules out a common driver. That is wrong. The shift predictor does not rule out a common driver but it can only control for common effects on two neurons caused by the onset of a stimulus or repetitive task events.

This is a valid point and we now rephrased the statement to match the Reviewer's more accurate wording (page 8).

- Why did the authors only use trials $n \pm 1$ to compute the shift predictor? Most studies simply take all trial combinations. Does that make a difference in the analysis? If so, it could be related to non-stationarities in the signal.

We adopted the calculation of the shift predictor as implemented in the FieldTrip toolbox (ft_spike_xcorr.m), which we also used for most other analyses. By using the two most nearby trials to calculate the shift predictor, effects of non-stationarity are arguably optimally mitigated and render the shift-predictor a particularly stringent control.

6) In the example neuron in Fig. 3b there is higher activity in the HIPP cell during AM+ than NAM encoding. Was that generally the case?

Please note that the data in Figure 3b (now Figure 2d) show retrieval effects, not encoding.

- Were there general differences in HIPP and EC during encoding between AM+ and AM- trials? I.e. did higher activity predict better encoding?

Subsequent memory effects at encoding are in general more subtle than retrieval success effects, owing in part to the more indirect link between encoding activation and later memory success (including impacts of the delay period and of the retrieval process itself). In our study, this is exacerbated by the fairly low number of AM- trials (17 trials on average). Consequently, the encoding effects of AM+ vs. AM- are quite modest in our data. In HC, there is numerically greater activation for AM+ than AM- when averaging across the 0-3 s encoding period ($t_{(169)} = 1.42$, $P = .156$). The same holds true for EC ($t_{(167)} = 1.12$, $P = .263$). We thus prefer not to include the encoding data and also (in light of a later comment) remove the encoding portion from previous Figure S4, now keeping the memory effects focused on the retrieval phase.

- In Figure S4 there is an intriguing effect of an overall elevation of the correlation function in the AM+ trials during encoding (the left plot). This presumably means that the firing rates were better correlated across trials, without fine temporal structure (see e.g. Bair et al. J. Neurosci 2001 for an excellent explanation of how this works). However, whether this effect can be observed in the analysis depends on the details on the cross-correlation analysis (different labs make different choices regarding normalization and subtraction of the baseline), details which should be added to the Ms. Do the authors indeed observed such an increase in the general across-trial correlation on AM+ trials?

Based on the statistical pattern of the data shown in Figure S4, we initially wanted to make the point that rather than primarily firing *after* HC spikes as during retrieval, EC neurons also fire *before* HC spikes during encoding, consistent with a differential emphasis of EC-HC input at encoding. However, in light of the above statement regarding the relative weakness of the encoding effects in this study, we now removed Figure S4 from the manuscript. That said, we certainly agree with Reviewer 1 that exploration of single unit encoding effects would be a great endeavour for future studies.

7) In Fig. 4 there are differences between the visual stimuli (presence of an occluder) that could explain the difference between AM and NAM activity. This is not so likely given the timing of the effect on firing rate, but it should nevertheless be briefly discussed.

The occluder was included to optimize the reinstatement analysis for AM trials, i.e., holding everything except presentation of the object per se constant between encoding and retrieval. But the Reviewer is right, for the comparison of AM+ and NAM HIT trials, the visual occluder is another factor that differs between the two trial types (in addition to the main factor of interest, i.e., associative vs. non-associative retrieval processes). We agree though that the presence vs. absence of a gray square is unlikely to account for the classification difference >500 ms after stimulus onset. In particular, Figure 2c (previously Figure S1) indicates that firing rates increase for AM+ relative to AM- in the same latency range, with the only difference between these two trial types being the success vs. failure to deploy associative retrieval processes (and with the occluder being present in both cases). We now include mention of this potential confound in the Results (page 9).

- In the AM trials, the authors used 3s presentation and in the NAM trials 2s. Is this a confound? If not, please explain why.

The Reviewer is correct, AM trials lasted 1s longer than NAM trials. This was done (i) to avoid overtaxing patients given the increase in task difficulty for AM and (ii) to bring up performance for AM trials to better match NAM trials. We restricted our analyses to the first 2 seconds in both tasks when directly comparing NAM and AM trials. For AM-only analyses (reinstatement and cross-correlations), we now extend the analysis window to encompass the entire 3 seconds. Note though that the results are almost identical to using only the first 2 seconds.

8) The patients make a reasonable number of errors during the AM task, do the EC classifiers predict the incorrect object during these error trials?

This is another great suggestion. We now included direct comparison of EC reinstatement during AM+ vs. AM- trials. Indeed, results revealed significantly stronger reinstatement for AM+ trials, corroborating the behavioural relevance of the reinstatement effects (now shown in Figure 3b). In addition, we show the unthresholded reinstatement map for AM- trials in Figure S3a. In line with the Reviewer's intuition, the classifiers tended to predict the incorrect object (blue patches), but the trial numbers for AM- were still slightly too low in this paradigm for the effect to survive our stringent statistical criteria. We now also mention the negative classification (reinstatement of the non-target object) in the Discussion on page 12.

9) What happens to HIPP and EC activity during the 30s delay? Do the neurons code for the last presented item?

This is an intriguing idea, but the study was not designed to adequately address this question. First, we did not systematically control the content of the last item. More importantly, the 30s delay was not well controlled, e.g. by having participants perform a low-level secondary task. Instead, the experimenter occasionally engaged in casual conversation with the participant. Having said that, we recently showed via fMRI that the most recent items only 'echo' for a few seconds in EC and then EC seemed to reactivate previously encoded items to strengthen their mnemonic representation (Staresina et al., PNAS 2013).

10) In every trial, there were two encoding epochs and two decoding epochs. I suppose that the data were collapsed across epochs for the analysis. Were the effects also visible when the first and the second epochs were evaluated separately? Were there any differences between the epochs?

We assume the Reviewer is referring to Figure 1, where we only show two example trials for encoding and two example trials for retrieval. We apologize for the confusion, but the encoding phases consisted of 20 trials for NAM and 10 trials for AM, and the retrieval phases of 40 trials for NAM (20 old, 20 new) and 10 trials for AM (same as encoding, but without the object). We now expanded the legend for Figure 1 to better describe the trial structure and inserted '...' symbols to indicate that only 2 examples are shown in Figure 1.

11) The authors generally used a decoding approach. How good were these effects visible when the tuning of individual neurons was used instead?

This is a very good point which resonates with a suggestion made by Reviewer 3. In short, the reinstatement effects from our decoding approach are nicely mirrored when using the tuning of individual neurons. Specifically, we now added a new analysis in which we plot each EC cell's encoding effect of object 1 vs. object 2 (e.g., scorpions vs. raspberries) against the same cell's corresponding retrieval effect during AM+ trials. Results first show a significant positive correlation (Spearman $r = .19$, $P = .006$). Reflecting the timing of reinstatement observed with our decoding approach, this correlation was even stronger when only considering the firing rates from 1-1.6 s (Spearman $r = .28$, $P < .001$). Moreover, no reliable positive correlation was observed when considering AM- trials (Spearman $r = .02$, $P = .787$). As these results corroborate the decoding approach in a perhaps more traditional tuning framework, we now include them in the main text (Figure 3c) and in the supplement (Figure S3b). However, we want to emphasize that the decoding approach is arguably more compelling, as it uses second level statistics across participants rather than fixed-effect analyses across neurons.

Small points:

- Fig. 2e: explain the meaning of the black and grey bars

The black bars were the mean count of the normalised firing rates, with the grey bars denoting the standard error of the mean. We now switched to the representation in the inset and explain in more detail in the figure legend what the solid lines and shaded areas refer to.

- Fig. 4c: can you make the scale on the x-axis and y-axis similar? That would provide a more realistic impression of the degree of tuning for the two factors

Good point, we now equated the axes in all scatter plots. Note also that we now excluded NAM CR trials from the analysis, as this allowed us to use exactly the same set of trials for both contrasts (i.e., (i) NAM HIT vs. AM+, pooled across buildings and landscapes and (ii) buildings vs. landscapes, pooled across NAM HIT vs. AM+).

- Analysis: explain the meaning of "baseline corrected". Corrected how?

For depictions of firing rates across time (Figure 2b-d, Figure 3d, Figure 5c), we subtracted, for each neuron, the average firing rate from the -.5 s to 0 s prestimulus baseline interval. Note though that the patterns were statistically identical for uncorrected time courses (see figure below) – the subtraction was merely done to better depict the post-stimulus increase in firing rates. We now describe this procedure in more detail in the Methods section (page 14).

Figure: Data from Figure 2b, shown with baseline correction (left) and without (right). Note that the three conditions differ in the same time window irrespective of baseline correction.

Reviewer #2 (Remarks to the Author):

The paper by Staresina et al uses single unit recording data from epileptic patients to look at the dynamics of neural activity in the hippocampus and entorhinal cortex during associative memory recall, and compares the activity with a simpler recognition task. This sort of research remains rare, because only a relatively small number of research centers are able to do this sort of investigation. But here, the authors add a number of twists that mean that the results are particularly intriguing and novel.

Essentially, they describe prolonged activity in the hippocampus on trials where the patient is required to remember whether a particular object was associated with a particular scene. Furthermore, this increased hippocampal activity is associated with selective activation of the entorhinal cortex (EC) – and thus may be perhaps the first clear demonstration of the action of a hippocampal – EC circuits in episodic memory. The study is particularly unusual in using cross correlations between pairs of hippocampal and EC neurons to provide evidence for linkage between the populations of neurons.

Given the novelty of the results and their potential implications for the understanding of the neural mechanisms underlying episodic memory, I am essentially quite enthusiastic about this paper, which is also well written.

We thank Reviewer 2 for their positive assessment of our paper

I nevertheless have some questions.

1) The experiments rely on having neurons that have already been demonstrated to have selectivity for particular objects – for example, scorpion vs raspberries. However, there is no description of how this selectivity was demonstrated. In particular, when they pair a particular object with an image, and then test with a pair of names (“scorpion” vs “raspberries”), is it the case that there are always neurons that are individually selective to both the test items? If the patient had been previously tested with lots of images of scorpions, but not with raspberries, this would unbalance the protocol. Can the authors confirm that all the test images are equally familiar?

This is a very valid concern, but as the Reviewer suggests, all object stimuli were equally familiar to the patients. Specifically, we chose 2 images from a larger pool of 100 images, each of which was presented 6 times during a previous screening session (now described in more detail in the Methods section, page 13-14). As for stimulus selectivity, there was not always a selective response to two images in two EC cells, which would allow for more examples like the one shown in Figure 3d. That said, we now also include an encoding-retrieval reinstatement analysis based on tuning profiles across EC neurons, which strongly corroborates the reinstatement findings from our population decoding approach (Figures 3c and S3b).

2) In the legend to figure 2b, it is stated that it plots “Mean hippocampal firing rates”, but since the firing rate before the stimulus comes on is zero, I presume that the firing rates have been baseline corrected. The responses look quite neat, but when you see that we are talking about a

maximal increase of 0.7 Hz across the population, this seems to be fairly weak. I presume that some individual neurons must have substantially larger increases in firing rate to achieve this. Some additional information about the reliability of such responses would be appreciated.

This is another good point. First, the Reviewer is correct in that we baseline-corrected the firing rate time courses and that on average (across the 238 neurons), the firing rate increase is .7 Hz. His/her intuition about some individual neurons having substantially larger increases in firing rate is also correct, as can be appreciated in the example HIPP neuron in Figure 2d where there is a ~16 Hz increase in firing rate. We present an overview of the distribution in pre- to post-stimulus firing rate changes below. Importantly though, the statistics for Figure 2b were done with neurons as the units of observation. That is, we compare firing rates for condition A vs. condition B within each neuron and then statistically test for the reliability of potential differences across neurons. In other words, while the mean firing rates in Figure 2b include differential contributions from different neurons, the statistical effects of interest are reliable across neurons.

Figure: Distribution of firing increases from baseline during AM+ across cells. For the distribution, we only included neurons that showed an increase from baseline (181 out of 238 neurons). *Top*: The majority of neurons show increases between 1 and 4 Hz. Red arrow highlights the neuron picked as an example in Figure 2d. *Bottom*: Distribution of time points of maximum firing increase, showing that the majority of maxima occur within the first second after stimulus onset.

Figure: Data from Figure 2b, shown with baseline correction (left) and without (right). Note that the three conditions differ in the same time window irrespective of baseline correction.

3) In the methods, the authors say that the cross-correlelogram data was obtained using all within-hemisphere combinations of HIPP/EC neuron pairs in a given participant (resulting in a total of 1798 combinations/histogram). Could the authors say how many hippocampal and EC neurons were involved?

Good point. Each participant provided an average of 112 (SEM = 35) EC/hippocampus combinations, with an average of 12 (SEM = 2) EC neurons and an average of 12 (SEM = 3) hippocampal neurons contributing. This is now stated in the Methods Section on page 15. Note though that only within-hemisphere combinations were used, such that the actual average number of combinations per participant (112) is smaller than $12 \times 12 = 144$.

Reviewer #3 (Remarks to the Author):

In this study, Staresita et al report on simultaneous single-neuron recordings performed in the human hippocampus and entorhinal cortex (EC) during two types of memory retrieval tasks. They report that i) hippocampal cells responded more during successful recall compared to successful recognition, ii) the identity of the recalled item could be decoded from EC, and iv) cross-correlation analysis indicates that hippocampal activity precedes that in EC during successful recall. The scientific novelty of this paper is principally the EC results.

The work is based on rarely available high-quality data and the experiments are performed well. The key novel technical aspects of this study are comparison of the same neurons in two tasks (recognition, recall) and the use of simultaneous EC-Hippo pairs of cells to study recall. This result is potentially of wide interest to the community, given the unique ability to study memory recall at single-neuron level in humans.

The results are interesting and the data of high quality. However, I am concerned about the robustness of the results and the way they are presented that limit my enthusiasm. The principle results on the EC (i.e. Fig 2d) are of small size and significance and not supported by single-neuron analysis or correlations with behavior. In contrast, the hippocampal results appear substantially more robust (Fig 4). It is thus unclear how robust the reported effects in EC are and whether they are of relevance for memory. Also, the writing is unnecessarily strong and does not properly relate these findings to earlier work in several instances. In conclusion, while interesting, I do not find that there is sufficiently robust support for the strong claims made in this study to warrant publication in this form. Having said that, an extensively revised version might be able to overcome these concerns.

We thank the Reviewer for his/her thorough and fair assessment of the study. In brief, we were able to address all concerns. First, we present the EC assembly decoding data in a different way (encoding time x retrieval time resolved and compared against surrogates), illustrating that the effects are highly robust. Second, we now also show reinstatement across individual neurons, which nicely corroborates the assembly decoding approach. Lastly, we now provide a direct link with behaviour, showing that both the assembly decoding approach and the single neuron reinstatement are stronger for successful compared to unsuccessful recall. Accordingly, we re-structured the figures to dedicate more space to the EC findings (now Figures 3 and S3). Also, we made better reference to the existing literature and expanded our discussion to link our findings to earlier single unit work.

Major issues:

1. Robustness. Decoding accuracies are small (AUC values ~0.52). This in particular applies to Fig 2d, which is the key result. The impression of marginal significance is strengthened by an absence of exact p-values for the significant clusters. Overall it is not clear whether the statistical approach taken was appropriate.

We thank Reviewer 3 for this important comment. The relatively low decoding accuracies in Figure 2d resulted from two aspects of our analysis. First, the initial analysis included a time-by-time decoding approach, where every time point from encoding was used to train the classifier, and every time point at retrieval was used to test the classifier (resulting in an encoding time x retrieval time

reinstatement matrix). However, the resulting matrix was then averaged across all encoding time points to result in an integrated time course of reinstatement at retrieval. This averaging procedure pushed down the resulting accuracy values. In light of Reviewer 3's suggestion to elaborate on our EC results, we now present the entire encoding x retrieval classification matrix, which reveals a maximum decoding accuracy of 59.41%. This effect is very robust, reflected by the fact that it survives stringent statistical testing: For one, we always test two-sided (although decoding accuracies are often tested in a one-sided fashion for increases from chance). Also, we use a cluster-based permutation method (Maris and Oostenveld, *Journal of Neuroscience Methods* 2007) to control for multiple comparisons, in this case all 123,201 time x time combinations (351 time points for both encoding and retrieval resulting from a sliding window from -.5 to 3 sec in 10 ms steps). We now emphasize the effect sizes and corresponding P values in the results and in the figures. Second, we improved decoding performance by using larger smoothing windows, i.e. applying a running average of 500 ms on firing rates, rather than applying the 50 ms FWHM Gaussian smoothing kernel. Finally, we now statistically compare decoding performance not against the standard value of 50%, but against performance derived from surrogate data. In particular, for each participant, we shuffled the training labels 100 times and averaged the resulting decoding performance to give a data-driven value of chance classification. While this procedure has been argued to be the more appropriate statistical test for decoding (Grootswagers et al., *Journal of Cognitive Neuroscience* 2017), the final results are virtually identical compared to using the fixed value of 50% for comparison.

2. Absence of single-unit analysis for EC neurons and absence of comparable analysis for EC vs Hippo. Apart from one example (Fig 3), the principle effects here are not reported for individual neurons. The results on hippocampal neurons in Fig 2b are averaged across all neurons. That seems like an odd choice, given that there are a variety of types of hippocampal neurons (as can be seen in Fig 4). Something like Fig 4 is needed for EC. In particular, how many EC neurons show evidence of reactivation due to free recall? Do any hippocampal neurons show such a signal? Also, some analysis is shown for one area (Hippo, Fig 2b) but not the other (is this also the case for EC?). This limits ability to compare results between areas.

We agree with Reviewer 3 that more single-unit results are informative. As suggested, we now added a single-unit reinstatement analysis for EC, plotting each unit's encoding effect (object 1 vs. object 2) against its corresponding effect at retrieval. Irrespective of the time windows chosen (e.g. both encoding and retrieval firing averaged across the whole trial or only in time windows where the decoding approach was significant), we observed a significant positive correlation. That is, cells that had a preference during encoding for object 1 also had a preference for object 1 during successful recall. We do think though that our second-level assembly decoding approach is a particular strength of our study. In particular, due to the rarity of these data, human single unit studies tend to suffer from relatively small sample sizes and lack of opportunity for replication. Hence we deem it especially important to conduct random effect analyses that strengthen conclusions at the population level, as fixed-effect analyses (pooling across patients and conducting statistics across units) may be disproportionately driven by a single patient. We now explicitly state this rationale in the Discussion (page 11-12).

We also followed Reviewer 3's suggestion and present corresponding results for both regions. As for reinstatement, we mention that we found no reinstatement effects in HIPPO. The univariate memory results for EC are now presented in Figure S1 and further discussed in the Discussion section (page 11).

3. Lack of citation of prior work and unfounded claims. Aspects of the work here have been shown before, in particular recollection-triggered activation of selective cells in entorhinal cortex and hippocampus (Gelbard-Sagiv et al. 2008 Science), selectivity of MTL cells to categories and objects (Fried 1997, Kreiman et al. 2000, Nat Neurosci, Quiroga et al 2005 Nature), and selectivity of MTL cells to familiarity/recollection. Also, there is work on simultaneous EC-hippo recordings in rodents addressing issues related to reinstatement. None of these are cited/discussed. A second writing issue is that unwarranted statements of causality are made, none of which can be substantiated without causal intervention. For example: " our findings reveal that neurons in the human hippocampus exert a driving influence on EC target sites ". In general, the scholarship of this paper needs to be improved.

We agree with Reviewer 3 that no claims about causality can be made without intervention. Accordingly, we removed all mention of causality and instead emphasize the temporal and correlative relationship between HIPP firing and EC reinstatement. We also explicitly acknowledge that no inference on causality can be made without experimental intervention (page 8).

As for related work, we take the Reviewer's point that related human single unit studies deserve more discussion. All mentioned studies are now cited (page 11), along with an explanation of how our current approach is quite different from them. While traditionally the number of units selectively responding to one or more stimuli was quantified across regions, we here examine assembly decoding performance for two particular object exemplars. Regarding familiarity/recollection in human MTL, we now additionally cite Rutishauser et al. (Neuron 2006), Viskontas et al. (Journal of Cognitive Neuroscience 2006) and Rutishauser et al. (PNAS 2008) in the Discussion (page 11). With regard to rodent work, however, we are not aware of any studies investigating HIPP-EC dynamics beyond the two we already cited (Ólafsdóttir et al., Nature Neuroscience 2016 and O'Neill et al., Science 2017). Also, as mentioned in the introduction, it is somewhat challenging to infer from rodent navigation to the phenomenological quality of human recollection.

4. Potential confound for cross-correlation analysis between simultaneously recorded EC-Hippo cell pairs. I am uncertain whether this analysis does not suffer from the following confound. As Fig 3 shows, neurons in hippocampus respond with a latency earlier than that in EC. Thus, there is a common input, but with a latency offset. It is unclear whether this is what drives the directional effects seen in the cross-correlation analysis (rather than a true spike-by-spike directional influence).

This is a valid concern, but two aspects of our analysis rule out this confound. First, potential latency differences from a common input are exactly what the shift-predictor is designed to mitigate. Imagine the scenario that HIPP simply responds faster to stimulus presentation than EC, e.g. first firing after 200 ms with EC first firing after 210 ms. As the Reviewer correctly states, this would produce the same pattern we showed in Figure 2e (now Figure 4b), with HIPP firing seemingly preceding EC firing. Critically though, what the shift predictor does is calculate the same cross-correlogram (CCG) between HIPP and EC, but with taking the HIPP spikes from trial n and the EC spikes from trial $n+1$. The hard-wired latency differences should be preserved across trials, such that the shift-predictor CCG would look identical to the actual same-trial CCG. Thus, one subtracts the shift-predictor CCG

from the same-trial CCG to tease out firing relationships that cannot be accounted for by a common input with different latencies.

Second, the same hard-wired latency differences between HIPP and EC should be present during AM- trials. However, as shown in Figure 4b, no difference from the shift-predictor baseline were observed for AM- trials.

5. Object identity encoding in EC. Object identity could be decoded from Hippocampus, but not EC. But this is not what the prior literature shows? This statement is based on what is shown in Fig S2. But this figure is far from convincing due to marginal differences. How many neurons encode objects in each area? Also it is at odds with the literature, which shows object identity coding in both EC and hippocampus (i.e. Mormann et al 2008 J Neurosci).

We assume there was a mix-up and Reviewer 3 actually points out the lack of object decodability in HIPP, not in EC. It is important to note that our decoding approach relies on patterns of activity across cell assemblies, rather than binary stimulus selectivity of single units. That said, we now quantified how many EC units and how many HIPP units show stronger firing rates for one of the two object images over the other. Specifically, we averaged the firing rates for the entire 3 sec of an encoding trial and compared, unit by unit, the firing rates for the 40 encoding trials showing object 1 with the 40 encoding trials showing object 2 (independent samples t test, $P < .05$, two-tailed). Results showed that an average of 1.2 HIPP units per participant (SEM = 0.6) and 2.2 EC units per participant (SEM = 0.5) showed statistically different response profiles for one object over the other. While this difference only shows a statistical trend ($t_{(15)} = 1.78$, $P = .096$), this univariate effect will still impact multivariate decodability as there are on average more diagnostic cells in EC than HIPP. This is now reported in the methods section on page 14 and we consequently now refrain from directly comparing decodability of objects in EC vs. HIPP. Note though that the reinstatement effects in EC (now Figures 3b and S3) are not biased in any fashion, as the key finding here is that EC response patterns from perceptual encoding return upon recall in the absence of visual object information. Finally, the finding of fewer object-selective responses in HIPP than EC is consistent with a recent study by Mormann et al. (Nature Neuroscience 2011), showing that objects are underrepresented in the hippocampus compared to EC. This is now mentioned in the Discussion on page 11.

6. Was the extent of reinstatement in EC indicative of successful recall? More extensive support for the very interesting and novel result of a potential role of reinstatement in EC for memory recall would strengthen the paper.

Excellent point. The answer is yes, the extent of reinstatement in EC was indicative of successful recall. We directly contrasted the encoding x retrieval reinstatement maps for AM+ vs. AM- and found that reinstatement of ~1000-1500 ms encoding pattern at 500-1500 ms during retrieval was significantly stronger for successful recall, again surviving stringent correction for multiple comparisons. This is now shown in Figure 3b. Likewise, unlike for AM+ trials, AM- trials did not show a positive encoding-retrieval correlation of effect sizes across neurons, shown in Figure S3b. These are both key results to support a link between EC reinstatement and memory behaviour.

Minor:

1. In some plots the units are confusing. For example, in Figs 3-4, there are negative firing rates?

We apologize for the confusion. Negative firing rates result from baseline correction, i.e., there was a relative suppression in firing rates after stimulus onset. We think that showing baseline-corrected data is still informative, but note that the peri-stimulus time histograms are not baseline-corrected. We now explicitly describe the baseline correction in the Methods on page 14.

2. All statistical claims need to be accompanied by an effect size and p-value

We now added the exact P values for the clusters as well as the t/F statistic in its corresponding P value for each maximum to both text and figures.

3. Why did the decoder in Fig 2d achieve what appears to be significant below-chance performance shortly after stim onset?

The non-significant below-chance values were merely a result of noise fluctuations. In the revised manuscript, we improved our decoding analyses in two ways. First, rather than using the time series resulting from convolution of the spike trains with a Gaussian kernel, we now average the spike trains across 500 ms sliding windows. This results in more stable and robust features for classification while still providing good temporal resolution. Second, rather than simply comparing decoding performance against chance level (50%), we now derive a surrogate decoder via randomly shuffling the training labels 100 times (Grootswagers et al., Journal of Cognitive Neuroscience 2017). The actual decoding performance is then compared against the surrogate decoding performance. Note though that results are virtually identical to testing against the 50% chance value.

REVIEWERS' COMMENTS:

Reviewer #1 (Remarks to the Author):

Re-review of Staresina et al

The authors did an excellent job in addressing my concerns and I recommend publication.

They should address a few remaining points:

1) P. 4: "However, if hippocampal firing rates were merely driven by task difficulty, one might have expected a further increase for AM- trials, which are arguably more difficult than AM+ trials. We thus suggest that the increase in hippocampal firing rates from 500 ms onward reflects successful deployment of associative retrieval processes."

- That is an odd conclusion. If a subject makes an error, that is not evidence for a higher difficulty. He or she could simply be thinking about something else

2) P. 6: "Results showed decoding performance significantly exceeding chance by 12.94% (SEM=3.58%) across participants ($t(15) = 3.62$, $P = .003$ vs. surrogates)." The value of 13% is much higher than the values reported in Fig. 4a, please explain the difference.

3) P. 9 In the section on "Early scene coding followed by associative retrieval in hippocampus" it would be useful to mention the window size.

- Fig. 5a,b what is the x-axis? The centre of the computational window?

4) P. 9 Can you spell out the meaning of the "Time Window x Decodability" interaction, i.e. how one should interpret it? Many readers will not have followed the purpose of testing the interaction.

5) P. 9 There is no Fig. 3e.

6) P. 9 "As shown in Figure 5c, there was no significant correlation between the two effects across neurons (Spearman $r = .11$, $P = .105$), suggesting that hippocampal scene-discrimination vs. associative retrieval effects are not only temporally dissociated, but largely carried by separate neuron populations. This result is reminiscent of 'visually-selective' vs. 'memory-selective' neurons in human MTL recordings reported previously." This conclusion is wrong for two reasons. First, it is not appropriate to draw such a strong conclusion from a non-significant test. Second, even if the correlation would be exactly zero, one cannot conclude that there are separate populations. That conclusion would only be appropriate in case of a substantial negative correlation, or some other measure of a dichotomous population.

7) P. 11 The authors suggest that hippocampal neurons might provide indices to specific memory episodes. How does that suggestion relate to the finding of reproducible decoding of scene types across trials? Does that not imply tuning for scenes rather than an index to a particular episode?

Reviewer #2 (Remarks to the Author):

I have read the responses of the authors to my original review, and am satisfied by their replies. I also think that they appear to have done a serious job in replying to the concerns of the other two reviewers.

Reviewer #3 (Remarks to the Author):

The authors are presenting an extensively revised version of their manuscript. I commend the

authors for a very strong revision. I note in particular the strong novel result of behavioral relevance by comparing AM+ vs AM- in EC, now demonstrating relevance of reinstatement for success of retrieval. Together, this high-quality dataset and analysis is an important new contribution to the literature on human memory. Also I find the discussion much improved.

All my major points are addressed and I strongly support publication in this form.

Minor points:

1. Re Reinstatement (Discussion point): It might be worth commenting on the TCM model (Howard & Kahana 2002), which makes detailed predictions on the form of such reinstatement during recollection. Also, experimentally, one single-neuron study that comes to mind that shows evidence for reinstatement during verbal cued recall in human MTG is Jang et al 2017 Current Biology – it might be worth relating to this in discussion.

2. Some of the fonts are too small, for example the labels on the rasters in Fig 3 bottom row. Please adjust. Also, Fig 4a is missing a color bar for the 2D plot.

Responses to Reviewer Requests

Reviewer #1

Re-review of Staesina et al

The authors did an excellent job in addressing my concerns and I recommend publication. They should address a few remaining points:

1) P. 4: “However, if hippocampal firing rates were merely driven by task difficulty, one might have expected a further increase for AM- trials, which are arguably more difficult than AM+ trials. We thus suggest that the increase in hippocampal firing rates from 500 ms onward reflects successful deployment of associative retrieval processes.”

- That is an odd conclusion. If a subject makes an error, that is not evidence for a higher difficulty. He or she could simply be thinking about something else.

- This is a fair point, and we now removed the corresponding sentences from the manuscript.

2) P. 6: “Results showed decoding performance significantly exceeding chance by 12.94% (SEM=3.58%) across participants ($t(15) = 3.62$, $P = .003$ vs. surrogates).” The value of 13% is much higher than the values reported in Fig. 4a, please explain the difference.

- We apologise for the confusion, but the quoted section refers to object decodability during the study phase and is depicted in supplemental figure 2. The data in main figure 4a relate to decodability of the target object during retrieval. The bar graph values are slightly smaller than the maximum accuracy value reported in Figure 3b (~51%/~54% vs. 59.41%) because they reflect the average within the black rectangle, i.e. accuracy values collapsing across the 1-2 sec encoding window and the .6-1.5 sec retrieval window.

3) P. 9 In the section on “Early scene coding followed by associative retrieval in hippocampus” it would be useful to mention the window size.

- Thank you, we now added the information “averaging firing rates across a sliding 500 ms window” to the corresponding section on page 9.

- Fig. 5a,b what is the x-axis? The centre of the computational window?

- Yes. We added “x-axis reflects decoding time points after averaging firing rates in a +/- 250 ms window” to the figure legend.

4) P. 9 Can you spell out the meaning of the “Time Window x Decodability” interaction, i.e. how one should interpret it? Many readers will not have followed the purpose of testing the interaction.

- Thank you for the good suggestion. We now added the sentence “This interaction emphasises the relative increase in scene type decodability early in the trial and the relative increase in retrieval task decodability later in the trial.”

5) P. 9 There is no Fig. 3e.

- Thank you, this was meant to say Figure 3c and has now been corrected.

6) P. 9 “As shown in Figure 5c, there was no significant correlation between the two effects across neurons (Spearman $r = .11$, $P = .105$), suggesting that hippocampal scene-discrimination vs. associative retrieval effects are not only temporally dissociated, but largely carried by separate neuron populations. This result is reminiscent of ‘visually-selective’ vs. ‘memory-selective’ neurons in human MTL recordings reported previously.” This conclusion is wrong for two reasons. First, it is not appropriate to draw such a strong conclusion from a non-significant test. Second, even if the correlation would be exactly zero, one cannot conclude that there are separate populations. That conclusion would only be appropriate in case of a substantial negative correlation, or some other measure of a dichotomous population.

- This is a valid point. We rephrased that sentence to say “[...] suggesting that hippocampal scene-discrimination vs. associative retrieval effects are not only temporally dissociated, but are unlikely to be carried by the same neuron populations.”

7) P. 11 The authors suggest that hippocampal neurons might provide indices to specific memory episodes. How does that suggestion relate to the finding of reproducible decoding of scene types across trials? Does that not imply tuning for scenes rather than an index to a particular episode?

- This is another excellent point. We would speculate that ‘visually-selective’ neurons, i.e., those neurons that represent buildings vs. landscapes in our paradigm, generalise across episodes. Conversely, ‘memory-selective’ neurons, i.e., those neurons that are preferentially involved in associative memory tasks, provide trial-unique episodic indices. Accordingly, we now added the following sentence to the discussion: “Nevertheless, specific semantic representations that are context-independent have likewise been found at the level of single hippocampal neurons³⁵, consistent with the present finding that hippocampal population codes allow decoding of two different scene categories (buildings vs. landscapes, Figure 5a) across trials. One interesting question for future research is whether firing patterns in ‘visually-selective’ neurons tend to generalise across episodes, whereas ‘memory-selective’ neurons, preferentially engaged in associative memory processes, provide trial-unique episodic indices.”

Reviewer #3

The authors are presenting an extensively revised version of their manuscript. I commend the authors for a very strong revision. I note in particular the strong novel result of behavioral

relevance by comparing AM+ vs AM- in EC, now demonstrating relevance of reinstatement for success of retrieval. Together, this high-quality dataset and analysis is an important new contribution to the literature on human memory. Also I find the discussion much improved.

All my major points are addressed and I strongly support publication in this form.

Minor points:

1. Re Reinstatement (Discussion point): It might be worth commenting on the TCM model (Howard & Kahana 2002), which makes detailed predictions on the form of such reinstatement during recollection. Also, experimentally, one single-neuron study that comes to mind that shows evidence for reinstatement during verbal cued recall in human MTG is Jang et al 2017 *Current Biology* – it might be worth relating to this in discussion.

- We thank the Reviewer for this good suggestion. Based on the Jang et al. paper, we added the following short paragraph to the discussion:

“Given its functional-anatomical proximity to the hippocampus, EC has been postulated as the first site of target reinstatement (Teyler and Rudy, 2007). That said, it is likely for successful recall to encompass reinstatement across multiple cortical modules as recollection unfolds. Indeed, a recent study recording from neurons in the human anterior temporal lobe has shown evidence for reinstatement of semantic information in this region during successful recall (Jang et al., 2017). Simultaneous recordings from lower-level perceptual and higher-level semantic regions would allow tracking the gradual emergence of a full-blown memory, alongside a possible reversal of information flow from encoding to retrieval (Linde-Domingo et al., 2019).”

- Regarding Howard and Kahana’s TCM model, we agree that this is a very elegant framework, but it is tailored specifically to account for behavioural patterns in free recall, elucidating the possible mechanisms governing endogenous generation of retrieval cues - rather than cued recall through exogenous retrieval cues as provided in our current study.

2. Some of the fonts are too small, for example the labels on the rasters in Fig 3 bottom row. Please adjust. Also, Fig 4a is missing a color bar for the 2D plot.

- Thank you, this has now been adjusted. We rearranged the panels in Figure 3 from landscape to portrait, making better use of space constraints. We also added the colour bar in Figure 4a.